# The Representation Landscape of Few-Shot Learning and Fine-Tuning in Large Language Models

**Diego Doimo**     **Alessandro Serra**     **Alessio Ansuini**     **Alberto Cazzaniga**

**Area Science Park, Trieste, Italy**
{diego.doimo,alessandro.serra,alessio.ansuini,alberto.cazzaniga}
@areasciencepark.it

## Abstract

In-context learning (ICL) and supervised fine-tuning (SFT) are two common strategies for improving the performance of modern large language models (LLMs) on specific tasks. Despite their different natures, these strategies often lead to comparable performance gains. However, little is known about whether they induce similar representations inside LLMs. We approach this problem by analyzing the probability landscape of their hidden representations in the two cases. More specifically, we compare how LLMs solve the same question-answering task, finding that ICL and SFT create very different internal structures, in both cases undergoing a sharp transition in the middle of the network. In the first half of the network, ICL shapes interpretable representations hierarchically organized according to their semantic content. In contrast, the probability landscape obtained with SFT is fuzzier and semantically mixed. In the second half of the model, the fine-tuned representations develop probability modes that better encode the identity of answers, while the landscape of ICL representations is characterized by less defined peaks. Our approach reveals the diverse computational strategies developed inside LLMs to solve the same task across different conditions, allowing us to make a step towards designing optimal methods to extract information from language models.

## 1 Introduction

With the rise of pre-trained large language models (LLMs), supervised fine-tuning (SFT) and in-context learning (ICL) have become the central paradigms for solving domain-specific language tasks [1]. Fine-tuning requires a set of labeled examples to adapt the pre-trained LLM to the target task by *modifying* all or part of the model parameters. ICL, on the other hand, is preferred when little or no supervised data is available. In ICL, the model receives an input request with a task description followed by a few examples. It then generates a response based on this context *without updating* its parameters.

While operationally, the differences between SFT and ICL are clear in how they handle model parameters, it is less clear how they affect the model's *representation space*. Although both methods can achieve similar performance, it is unknown whether they also structure their internal representations in the same way. Differently from recent contributions which compared ICL and SFT in terms of generalization performance [2–4] and efficiency [5], in this work we analyze how these two learning paradigms affect the *geometry* of the representations.

Previous studies described the geometry of the hidden layers using distances and angles [6, 7], often relying on low-dimensional projections of the data [8–10]. In contrast, we take a density-based approach [11] that leverages the low-dimensionality of the hidden layers [12–14] and finds

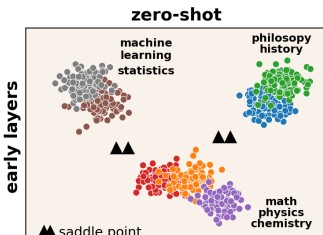 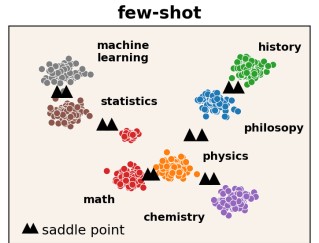 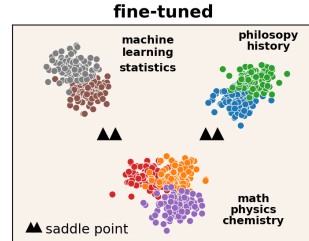

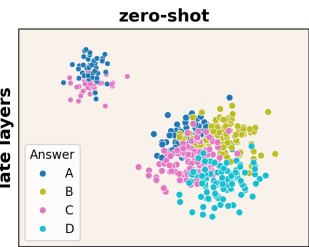 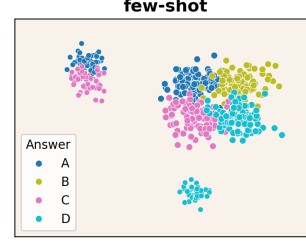 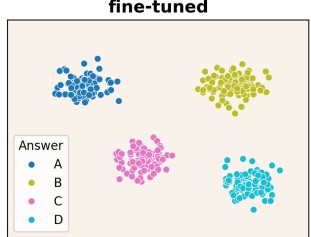

(a) **Consistency between the peak composition and the MMLU subjects.** A schematic view of the density peaks in the *early*. The coloring reflects the presence of the subjects.

(b) **Consistency between the peak composition and the MMLU answers.** A schematic view of the density peaks in the *late* layers. The coloring reflects the presence of answers with the same letter: A (blue), B (okra), C (pink), and D (light blue).

Figure 1: **The LLMs representation landscape of few-shot learning and fine-tuning.** This figure illustrates the distribution of probability modes in large language models (LLMs) during a question-answering task (MMLU). The top row shows representations from layers near the input, while the bottom row shows those near the output. We compare three scenarios: zero-shot (0-shot, left), in-context learning (5-shot, center), and fine-tuning (right). In the 5-shot scenario, early layers (top, center) develop better representations of the dataset's subjects. Conversely, in the fine-tuned model, the late layers (bottom, right) more accurately reflect better the letter answers.

the probability modes directly on the data manifold without performing an explicit dimensional reduction.

For ICL and fine-tuning, we track how the multimodal structure of the data evolves across different layers as LLMs solve a semantically rich multiple-choice question task and measure how the geometrical properties intertwine with the emergence of high-level, abstract concepts. We study ICL in a few-shot prompting setup and find that despite ICL and SFT can reach the same performance, they affect the geometry of the representations differently. Few-shot learning creates more interpretable representations in early layers of the network, organized according to the underlying semantics of data (Fig. 1, top-center). On the other hand, fine-tuning induces a multimodal structure coherent with the answer identity in the later layers of the network (Fig. 1, bottom-right).

The key findings of our study are:

1. Both few-shot learning and fine-tuning show a clear division between model layers, a marked peak in the intrinsic dimension of the dataset, and a sudden change in the geometrical structure of the representations (Secion 4.1);

2. Few-shot learning leads to semantically meaningful clustering of the representations in early layers, organized hierarchically by subject (Section 4.2);

3. Fine-tuning enhances the sharp emergence of clustered representations according to answers in the second half of the network (Section 4.3).

In summary, our geometric analysis reveals a transition between layers that encode high-level semantic information and those involved in generating answers. By studying the probability landscape on either side of this transition, we uncover how fine-tuning and few-shot learning take different approaches to extract information from LLMs, ultimately solving the same problem in distinct ways.

## 2 Related work

**Probing the geometry of the embeddings.** A classical approach to understanding the linguistic content encoded in token representations is probing [15]. Inspecting the embeddings with linear [16, 17] or nonlinear [18, 19] classifier probes allowed to extract morphological [20] syntactic [6] and semantic [17, 21–24] information. Probing has also been used to explore the geometrical properties of hidden layers. LLMs can represent linearly in hidden layers concepts such as space and time [25], the structure of the Othello game board [26], the truth or falsehood of factual statements [27]. The linear representation hypotheses [28] can be used to show that LLMs encode hyponym relations as simplices in their last hidden layer [29]. However, another line of work suggests that LLMs represent temporal concepts like days, months, and years, as well as arithmetic operations in a nonlinear way using circular and cylindric features [30, 31].

**Describing the geometry of the embeddings directly.** Directly analyzing the geometrical distribution of embedding vectors and their clustering can also provide insights into how LLMs organize their internal knowledge. Early studies on BERT, GPT2, and ELMo found that token embeddings are distributed anisotropically, forming narrow cones [32] and isolated clusters [7, 13]. This phenomenon has also been observed in CLIP embeddings [33], where image and text tokens occupy different cones, separated by a gap. A similar gap also exists between the subspace of different languages in multilingual LLMs [10], which enables approximating language translation with geometric translation between these subspaces. Recent work has shown that changes in the intrinsic dimension of the representations is related to different stages of information processing in LLMs, linking the rise of abstract semantic content to layers characterized by geometric compression [34] and the transition from surface-level to syntactic and semantic processing to layers of high dimension [35].

**Comparing in-context learning and fine-tuning in LLMs.** Recent studies have compared ICL to fine-tuning in LLMs by analyzing their ability to generalize, their efficiency, and how well they handle changes in the training data. Several factors can influence the outcomes when comparing ICL and fine-tuning. These include the format of the prompts in ICL [36, 37], the amount of training data used for fine-tuning [38, 39], and the size of models being compared [40]. In large models, ICL can be more robust to domain shifts and text perturbations than it is fine-tuning smaller-scale ones [2, 3]. However, when ICL and fine-tuning are compared in models of the same size fine-tuned on sufficient data, SFT can be more robust out-of-distribution, especially for medium-sized models [4]. Additionally, SFT achieves higher accuracy with lower inference costs [41].

## 3 Methods

### 3.1 Models and Dataset.

**MMLU Dataset.** We analyze the Massive Multitask Language Understanding question answering dataset, MMLU [42], taking the implementation of `cais_mmlu` from `Huggingface`. The MMLU test set is one of the most widely used benchmarks for testing factual knowledge in state-of-the-art LLMs [43–46]. The dataset consists of multiple-choice question-answer pairs divided into 57 subjects. All the questions have four possible options labeled with the letters "A," "B," "C," and "D." When prompted with a question and a set of options, the LLMs must output the letter of the right answer. In this work, we will analyze the MMLU test set, where each subject contains at least 100 samples, with a median population of 152 samples and the most populated class containing about 1534 examples. To reduce the class imbalance without excessively reducing the dataset size, we randomly choose up to 200 examples from each subject. The final size of our dataset is 9181.

**Language models and token representations analyzed.** We study the models of Llama3 [45], Llama2 [47] families, and Mistral [43]. We choose these LLMs because, as of May 2024, they are among the most competitive open-source models on the MMLU benchmark, with an accuracy significantly higher than the baseline of random guessing (25%, see Table A1). All the models we analyze are decoder-only, with a layer normalization at the beginning of each attention and MLP block. Llama2-7b, Llama3-8b, and Mistral-7b have 32 hidden representations, Llama2-13b 40, Llama2-70 and Llama3-70 80. In all cases, we analyze the representation of the *last token of the prompt* after the normalization layer at the beginning of each transformer block. For transformers

trained to predict the next token, the last token is the only one that can attend to all the sequence, and in the output layer, it encodes the answer to the question.

**Few-shot and fine-tuning details.** We sample the shots from the MMLU dev set, which has five examples per subject. This choice differs from the standard practice, where the shots are always given in the same order for every input question. Table A1 shows that the final accuracies are consistent with the values reported in the models' technical reports [43, 45, 47]. We fine-tune the models with LoRA [48] on a data set formed by the union of the dev set and some question-answer pairs of the validation set to reach an accuracy comparable to the 5-shot one. The specific training details are in Sec. A.

## 3.2 Density peaks clustering

We study the structure of the probability density of data representations with the Advanced Density Peaks algorithm (ADP) presented in D'Errico et al. [11] and implemented in the DADApy package [49]. ADP is a mode-seeking, density-based clustering algorithm that finds the modes of the probability density by harnessing the low-dimensional structure of the data without performing any explicit dimensional reduction. ADP also estimates the density of the saddle points between pairs of clusters, which measures their similarity and provides information on their hierarchical arrangement. At a high level, the ADP algorithm can be divided into three steps: the estimation of the data's *intrinsic dimension* (ID), the estimation of the *density* around each point, and a final *density-based clustering* of the data.

**Intrinsic dimension estimation.** We measure the ID of the token embeddings with Gride [50]. Gride estimates the ID of data points embedded in $\mathbb{R}^D$, using the distances between a token and its nearest neighbors. This is done by maximizing the likelihood function $L(\mu_k) = \frac{d(\mu_k^d - 1)^{k-1}}{B(k,k)\mu_k^{d(2k-1)+1}}$ where $\mu_k$ is the ratio of the Euclidean distances between a point and its nearest neighbors of rank $k_2 = 2k$ and $k_1 = k$, $d$ is the ID, and $B(k,k)$ a normalizing Beta function. By increasing the value of $k$, the ID is measured on nearest neighbors of increasing distance. The ID estimate is chosen as the value where the ID is less dependent on the hyperparameter $k$ and the graph $d(k)$ exhibits a plateau [50, 51]. On this basis, we choose $k = 16$.

**Density estimation.** We measure the local density $\rho_{i,k}$ with a $k$NN estimate: $\rho_{i,k} = {}^k/_{NV_{i,k}}$. Here, $N$ is the number of data points and $V_{i,k}$ the volume of the ball, which has a radius equal to the distance between the point $i$ and its $k^{th}$ nearest neighbor. Crucially, in this step, we compute the volume using the intrinsic dimension, setting $k = 16$, the value used to estimate the ID.

**Density-based clustering.** With the knowledge of the $\rho_i$, we find a collection of density maxima $\mathcal{C} = \{c^1, ...c^n\}$, assign the data points around them, and find the *saddle point density* $\rho^{\alpha,\beta}$ between a pair of clusters $c_\alpha$ $c_\beta$ with the procedure described in Sec. B. We can not regard all the local density maxima as genuine probability modes due to random density fluctuations arising from finite sampling. ADP assesses the statistical reliability of the density maxima with a $t$-test on $\log \rho^\alpha - \log \rho^{\alpha,\beta}$, where $\rho^\alpha$ is the maximum density in $c_\alpha$. Once the confidence level $Z$ is fixed, all the clusters that do not pass the $t$-test are merged since the value of their density peaks is compatible with the density of the saddle point. The process is repeated until all the peaks satisfy the $t$-test and are statistically robust with a confidence $Z$. We set $Z = 1.6$, the default value of the DADAPy package.

In the following sections, we will also use the notion of *core cluster points* defined as the set of points with a density higher than the lowest saddle point density. These are the points whose assignation to a cluster is considered reliable [11, 52]. With a slight abuse of terminology, we will use the terms "clusters", "density peaks", and "probability modes" interchangeably.

**Measuring the cluster similarity.** The ADP algorithm considers two clusters similar if connected through a high-density saddle point. This is done defining the *dissimilarity* $S_{\alpha,\beta}$ between a pair of clusters $c_\alpha$ and $c_\beta$ as $S_{\alpha,\beta} = \log \rho^{max} - \log \rho^{\alpha,\beta}$, $\rho^{max}$ being the density of the highest peak. With $S_{\alpha,\beta}$, we perform hierarchical clustering of the peaks. We link the peaks starting from the pair with the lowest dissimilarity according to $S_{\alpha,\beta}$ and update the saddle point density between their union $c_\gamma$ and

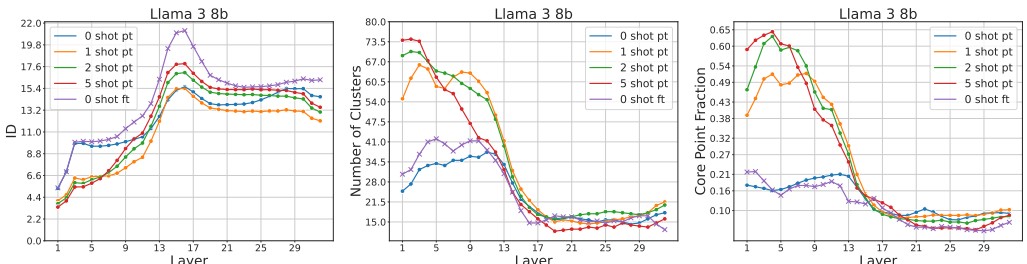

Figure 2: **Intrinsic dimension, number of density peaks, and fraction of core points.** Figure shows the ID (left), the number of density peaks (center), and the fraction of core points (right) for the last-token representation of Llama3-8b for an increasing number of few-shots and fine-tuned models. The three quantities change in the proximity of layer 17 in a two-phased fashion.

the rest of the peaks $c_{\delta_i}$ with the WPGMA linkage strategy [53]: $\log \rho^{\gamma,\delta_i} = \dfrac{\log \rho^{\alpha,\delta_i} + \log \rho^{\beta,\delta_i}}{2}$. After the update of $S$, we repeat the procedure until we merge all the clusters. In this way, we display the *topography* [11] of the representation landscape of a layer, namely the relations between the density peaks, as a dendrogram.

**Reproducibility.** We run the experiments on a single Nvidia A100 GPU with a VRAM of 40GB. Extracting the hidden representation of 70 billion parameter models requires 5 A100, and their fine-tuning requires 8 A100. We provide code to reproduce our experiments at https://github. com/diegodoimo/geometry_icl_finetuning.

## 4 Results

### 4.1 The geometry of LLMs' representations shows a two-phased behavior.

We start by exploring the geometric properties of the representation landscape of LLMs. Our analysis proceeds from a broad description of the manifold geometry to its finer details. First, we measure the intrinsic dimension (ID) to understand the global structure of the data manifold. Next, we will describe the intermediate-scale behavior, counting the number of probability modes on it. Finally, we analyze the density distribution at the level of individual data points within the clusters. These three quantities consistently show a two-phased behavior across the hidden layers of the LLMs we analyzed. All profiles of this and the following sections are smoothed using a moving average over two consecutive layers. We report the original profiles in the Appendix from Sec. D.2 to D.4 and in Sec. D.6.

**Abrupt changes in intrinsic dimension and probability landscape in middle layers.** We measure the ID of the hidden representations with the Gride algorithm (see Sec. 3.2). Figure 2 shows the results for Llama3-8b; the analysis on the other models can be found in Sec. D.2 to D.4 of the Appendix. The left panel shows that the ID changes through the layers with two phases, increasing during the first half of Llama3-8b and decreasing towards the output layers. Specifically, the ID rises from 2.5 after the first attention block and peaks around layer 16. The value at the peak increases with the number of shots, from 14 in the base model to 16.5 when a 5-shot context is added. The fine-tuned model (0-shot) reaches a maximum ID of 21 at this layer. In the second half of the network, the IDs sharply decrease over the next three layers. For the few-shot representations, the ID profiles gradually decay in the final part of the network, while for the 0-shot models, the ID increases again.

The same two-phased behavior appears in the evolution of the number of clusters on the hidden manifold (center panel). In the first half of the network, the probability landscape has a higher number of modes, ranging between 60 and 70, when the model is given shots, roughly matching the number of subjects. In the 0-shot and fine-tuned cases, it remains below 40. After layer 16, the number of peaks decreases significantly, remaining between 10 and 20. The right panel describes the representation landscape within individual clusters, measuring the fraction of core cluster points – those with a density higher than the lowest saddle point, indicating a reliable cluster assignment [11].

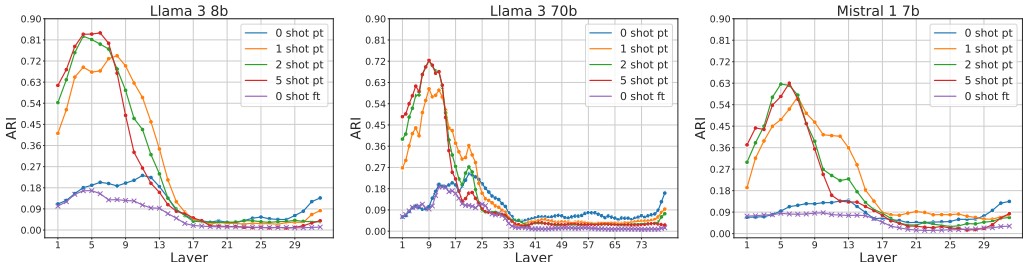

Figure 3: **Adjusted Rand Index (ARI) between clusters and subjects.** ARI between clusters and the subjects for Llama-3-8b (left), Llama-3-70b (center), and Mistral-7b (right) for an increasing number of few-shots and fine-tuned representations. In all cases, the match between cluster and subjects partition is highest at the beginning of the network and for an increasing number of shots.

Before layer 17, the core point fraction is higher, indicating a better separation between probability modes. Notably, the few-shot setting shows a core-point fraction of about 0.6, much higher than the 0-shot and fine-tuned case, which remains around 0.2.

The evolution of ID, number of clusters, and core cluster points is qualitatively consistent among the models we analyzed. In the Llama2 family, the ID peak is less evident (Fig. A3-bottom). In particular, it is absent for the less accurate model, Llama2-7b. Nonetheless, the number of clusters, core points, and the remaining quantities, presented in the following sections, change according to the same two-phased trend as the other models.

## 4.2 The probability landscape before the transition.

We now describe the data distribution by focusing on the semantic content of the last token, specifically analyzing whether the cluster composition is consistent with the prompt's topic. Using the 57 MMLU subjects as a reference, we compare the differences in the early layers of the LLMs between ICL and fine-tuning.

**Few-shot learning forms clusters grouped by subject.** To evaluate how well the clusters align with the subjects, we use the Adjusted Rand Index (ARI) [54]. An ARI of zero indicates that the density peaks do not correspond to subjects, while an ARI of one means a perfect match (see Appendix C for a detailed presentation). Figure 3 shows that as the number of few-shots increases, the ARI rises from below 0.27 in the 0-shot context to 0.82 in Llama3-8b (left), 0.72 in Llama3-70b (center) and 0.63 in Mistral (right) in the 5-shot settings. These ARI values correspond to a remarkable degree of purity of the clusters with respect to the subject composition. When the ARI is at its highest, 75 out of 77 clusters in Llama3-8b (layer 4), 53 out of 69 in Llama3-70b (layer 7), and 43 out of 53 in Mistral (layer 5) contain more than 80% of tokens from the same subject. In the next section, leveraging this high homogeneity of the clusters, we will connect the cluster similarity to the similarity between the subjects of the points they contain.

Additionally, when the number of shots grows, the ARI peak shifts to earlier layers, and the peak becomes narrower. For example, in Llama3-8b, the 0-shot profile (blue) has a broad plateau extending until layer 13. With one and two-shot settings (orange and green), the profiles show a couple of peaks between layers 3 and 9, and with 5-shots (red), a single large maximum at layer 3. The trend is consistent across other models, especially those with 70 billion parameters (see Fig. 3-center for Llama3-70b, and Fig. A9 in the Appendix for Llama2-70b). In all cases, providing a longer, contextually relevant prompt enables models to identify high-level semantic features (i.e., the subjects) more accurately and earlier in their hidden representations.

Few-shot prompting is not the only factor that increases the ARI with the subject. In the 0-shot setup, as the performance improves, LLMs organize their hidden representations more coherently with respect to the subject. In Llama3-8b and 70b models, where the 0-shot accuracy is above 62% (see Table A1), the 0-shot ARI is around 0.25. For the rest of the models with lower accuracy (below 56%), the ARI is below 0.18 (see blue profiles in Figs. 3 and A7).

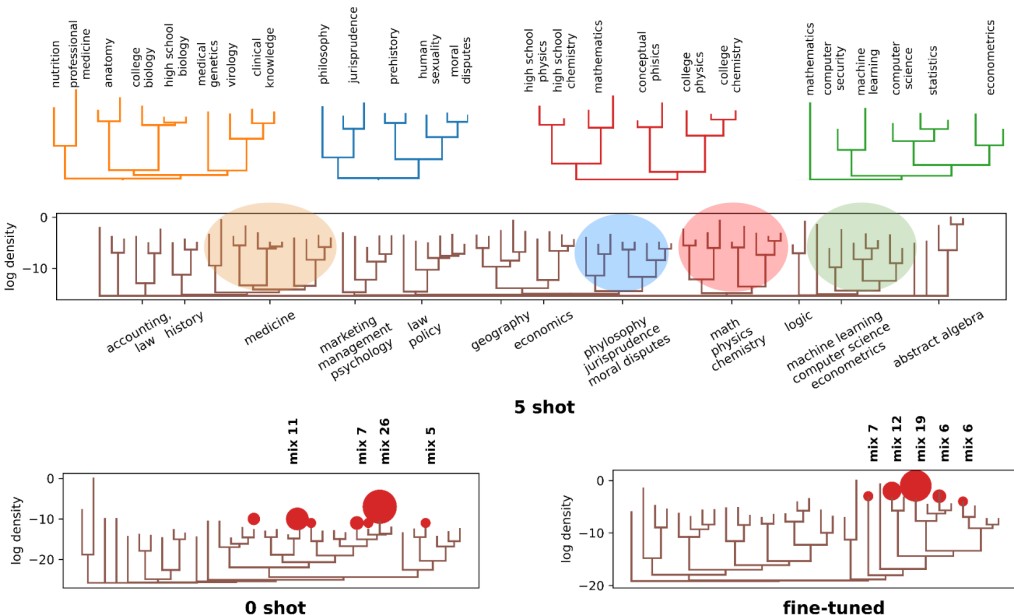

Figure 4: **Density peaks in the layers that best encode the subjects in Llama3-8b.** The dendrograms show the organization of the density peaks in Llama3-8b in the layers where the ARI with the subjects is highest for the 5-shot setup (top) and 0-shot set-up (bottom left) and fine-tuned model (bottom-right). In the 5-shot setup, the clusters are populated by examples from one or two related subjects, and their similarity reflects the semantic relationships between the subjects. In 0-shot and fine-tuned representations (bottom panels), some large clusters contain many subjects.

**The distribution of the density peaks mirrors the subject similarity.** When the models learn "in context", not only do the number and composition of density peaks become more consistent with the subjects, but they also organize hierarchically to reflect their semantic relationships. In this section, we describe only the cluster distribution in Llama3-8b in the layers where the ARI is highest and report the other models in the Appendix, Sec. D.7.

Figure 4 shows the probability landscapes of the Llama3-8b in 0-shot (bottom left), 5-shot (top), and fine-tuned model (bottom-right) as dendrograms. Dendrograms are helpful visual descriptions of hierarchical clustering algorithms [55]. We perform hierarchical clustering of the density peaks using the agglomerative procedure described in Sec. 3.2. In the layers where the ARI is highest, the density peaks are homogeneous, and we can assign a single subject to each leaf of the dendrogram. This one-to-one mapping between clusters and subjects allows us to estimate subject similarity based on the dendrogram obtained from the (density-based) hierarchical clustering of the peaks.

In all cases (0-shot, 5-shot, and fine-tuned model), clusters of subjects from the same broader field (STEM, medicine/biology, humanities, etc.) tend to be close together. However, in 0-shot and fine-tuned settings, the probability landscape has fewer and less pure density peaks at the subject level. In contrast, in the 5-shot setting, the number of clusters and their purity increase, and the 77 peaks are organized according to their high-level semantic relationships. For example, in the top panel of Fig. 4, four major groups of similar subjects can be identified: medicine and biology (orange), philosophy, jurisprudence, and moral disputes (blue), math, physics, and chemistry (red) and machine learning and computer science (green). In addition, these groups and hierarchical structures are consistent across different models (see A11 to A12). For instance, clusters related to statistics, machine learning, and computer science are often grouped together, as are those of chemistry, physics, and electrical engineering, or economy, geography, and global facts.

The structure we described is also robust to changes in the confidence level $Z$. In the Appendix, from Sec. E.2.1 to Sec. E.2.8, we report the dendrograms obtained with $Z = 0$ and $Z = 4$. Importantly, even with $Z = 0$, where all local density maxima are considered as probability modes, the probability landscape of the 0-shot and fine-tuned models remains largely mixed. In contrast, the probability landscape of 5-shot representation is more stable to variations of $Z$.

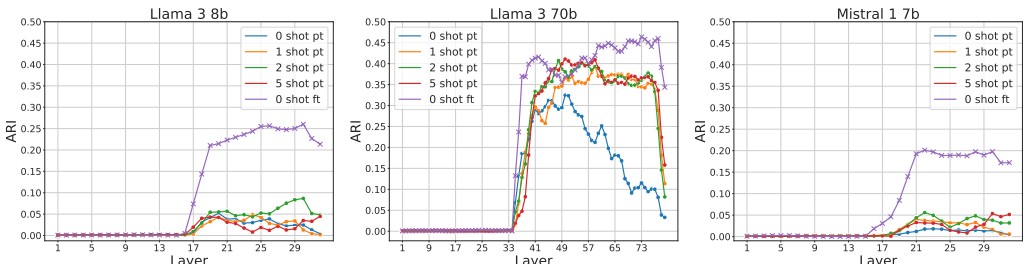

Figure 5: **Adjusted Rand Index between clusters and final answers.** Adjusted Rand Index (ARI) between clusters and the MMLU answers (test set) for Llama3-8b (left), Llama3-70b (center), and Mistral-7b (right). In the second part of the network, the purity of the clusters w.r.t the answer partition is highest for fine-tuned models.

## 4.3 The probability landscape after the transition.

In Sec. 4.1, we observed that the number of density peaks decreases in the middle layers of the network. This reduction happens because the model needs to identify the answer from four options at the output, causing points related to the same answer to cluster around the same unembedding vector. In addition, when the model is uncertain about its predictions, some output embeddings tend to lie close to the decision boundaries of the last hidden representation, resulting in a flatter density distribution with fewer peaks. Even when the model is highly accurate, the linear separability of the answers does not guarantee distinct density peaks because the embeddings may still be near the decision boundary as long as they are on the correct side. However, more pronounced density peaks emerge as the model confidence grows and the data moves away from the decision boundaries. This section shows that SFT sharpens these density peaks in the later layers more than ICL. However, as model size and accuracy increase, the representation landscapes of ICL and SFT become more similar.

**Fine-tuned density peaks better encode the answers better than few-shot ones.** We evaluate how well the clusters match the answer partition (i.e., "A," "B," "C," "D") using the ARI (see Sec. 4.2). When the models are fine-tuned, four to five large clusters emerge in the second part of the network, grouping answers with the same label. These clusters collect more than 70% of the data between layers 20 and 30 in Llama3-8b, more than 90% between layers 45 and 75 of Llama3-70b, and more than 65% between layers 21 and 30 in Mistral. In these clusters, the most common letter represents over 90% of the point in Llama3-8b, over 70% in Llama3-70b, and over 90% in Mistral. As a result, the ARI (see purple profiles in Fig. 5) rises sharply in the middle of the network, reaching approximately 0.25 in Llama3-8b, 0.45 in Llama3-70b, and 0.2 in Mistral. These ARI are related to the MMLU test accuracies of 65%, 78.5%, and 62%, respectively (see Table A2).

In contrast, in ICL, the clusters are more mixed, and their number is smaller. In Llama3-8b in the 5-shot setup, one cluster contains 70% of the points in the last layers. In the 0-shot case, four clusters with a roughly equal distribution of letters contain the same amount of data (blue profile). Similar trends appear in Mistral's late layers. In both models, the ARI values for few-shot context stay below 0.05 (Fig. 5). Interestingly, in Llama3-70b (and to a lesser extent in Llama2-70b, see Fig. A8-right), the representation landscape of ICL starts to resemble that of the fine-tuned models. Between layers 40 and 77, about 80% of the dataset forms five large peaks, and in four of them, the fraction of the most common letter is above 0.9, similar to fine-tuned models. Consequently, in these layers, the ARI for few-shot contexts (see orange, green, and red profiles in Fig. 5-center) oscillates between 0.35 and 0.40, except for the 0-shot profile, which decays from 0.3 to 0.05 (blue profile).

The different ways in which fine-tuning and ICL shape the representations of the network in the second half depend on the learning protocol, model size, and performance. In smaller models with moderate accuracy (below 65% / 70%), SFT and ICL can perform similarly as in the 5-shot setup (see Table A1) but they alter the geometry of the layers in different ways. However, with higher accuracy models like Llama3-70b (accuracy above 75%), both the performance of the model and the topography of the hidden representation tend to converge.

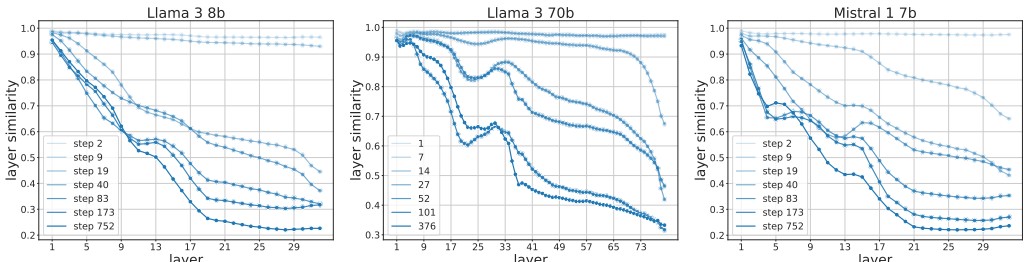

Figure 6: **Evolution of similarity between 0-shot and fine-tuned models during training.**. Panels show the dynamics of the representations' similarity with the base models for Llama3-8b (left) and Llama3-70b (center), and Mistral (right). Late representations change the most during fine-tuning.

**Fine-tuning primarily alters the representations after the transition.** In fine-tuned models, training leads to the emergence of structured representations that align with the labels. Figure 6 shows where, during the training, layers change the most in Llama3-8b (left), Llama3-70b (center), and Mistral (right). We compare the fine-tuned checkpoints with the 0-shot representations before training begins. To measure the similarity between representations, we use the neighborhood overlap metric [34, 56] that calculates the fraction of the first $k$ nearest neighbors of each point shared between pairs of representations, averaged over the dataset. Figure 6 shows that the similarity between representations is lower in the second part of the networks, decaying more sharply to its final value in the middle of the network from 0.5 to roughly 0.3 between layers 13 and 17 in Llama3-8b and Mistral, from 0.6 to 0.4 after layer 33 in Llama3-70b (see dark blue profiles).

This picture is consistent with what is shown in the previous sections. In the first half of the network, the representation landscapes of 0-shot and fine-tuned models are similar both geometrically (Fig. 2) and semantically (Figs. 3 and 4). In the second half of the network, where the representations are modified more during the training, fine-tuned models develop fewer peaks, more consistent with the label distribution than those of the other models (Fig. 5).

## 5   Discussion and conclusion

This study described how the probability landscape within the hidden layers of language models changes as they solve a question-answering task, comparing the differences between in-context learning and fine-tuning. We identified *two phases* in the model's internal processing, which are separated by significant changes in the geometry of the middle layers. The transition is marked by a peak in the ID and a sharp decrease in the number and separation of the probability modes. Notably, few-shot learning and fine-tuning display complementary behavior with respect to this transition. When examples are included in the prompt, the early layers of LLMs exhibit a well-defined hierarchical organization of the density peaks that recovers semantic relationships among questions' subjects. Conversely, fine-tuning primarily modifies the representations to encode the answers after the transition in the middle of the network.

**Advantages of the density-based clustering approach.** Our research highlights how variations in density within the hidden layers relate to the emergence of different levels of semantic abstraction, a concept previously explored by Doimo et al. [56] in convolutional neural networks (CNNs) trained for classification. In CNNs, the probability landscape remains unimodal until the last handful of layers, where multiple probability modes emerge according to a hierarchical structure that mirrors the similarity of the categories. In decoder-only LLMs solving a semantically rich question-answering task, these hierarchically organized density peaks appear in the early layers of the models, especially when they learn in context. Our methodology also extends the work of Park et al. [29], enabling the discovery of meaningful hierarchies of concepts *beyond* the final hidden layer of LLMs, where the data representation can be non-linear [30].

Moreover, unlike previous studies that utilized $k$-means to identify clusters within hidden representations [9, 13, 57], the density peak method does not assume a convex cluster shape or impose a priori the number of clusters. Instead, clusters emerge naturally once a specific Z value is set (see Sec. 3.2). This allows for the automatic discovery of potentially meaningful data categorizations based on the

structure of the representation landscape without specifying external linguistic labels and without introducing additional probing parameters. In this respect, an approach is similar to that of Michael et al. [58], who used a weakly supervised method relying on pairs of positive/negative samples to uncover latent ontologies within representations.

**Intrinsic dimension and information processing.** As discussed in section 4.1, a peak in intrinsic dimension separates two groups of layers serving different functions and being distinctly influenced by in-context learning and supervised fine-tuning. Other studies have also highlighted the crucial role of ID peaks in marking blocks of layers dedicated to different stages of information processing within deep neural networks. For example, Ansuini et al. [12] observed a peak of the ID in the intermediate layers of CNNs, separating layers that remove low-level image features like brightness from those that focus on extracting abstract concepts necessary for classification. In transformers trained to generate images, Valeriani et al. [34] identified two intermediate peaks delimiting layers rich in semantic features of the data characterized by geometric compression. In LLMs, Cheng et al. [35] showed that an ID peak marks a transition from representation that encodes surface-level linguistic properties to one rich in syntactic and semantic information. These studies suggest that ID peaks consistently indicate transitions between different stages of information processing within the hidden layers.

**Application to adaptive low-rank fine-tuning.** Our findings could improve strategies for adaptive low-rank fine-tuning. Several studies [59–61] tried to adjust the ranks of the LoRA matrices based on various criteria of 'importance' or relevance to downstream tasks. Our analysis of the similarity between fine-tuned and pre-trained layers (see Fig. 6) reveals that later layers are most impacted by fine-tuning, indicating that these layers should be assigned ranks. This approach would naturally prevent unnecessary modifications to the early layers during fine-tuning.

**Limitations.** Estimating the density reliably requires a good sampling of the probability landscape. This can be a delicate condition if the intrinsic dimension is high, as often happens in neural network hidden layers. The ID values we report in this work lie between 4 and 22, with most of the layers having an ID below 16. Rodriguez et al. [62] showed that the density can be estimated reliably up to 15-20 dimensional spaces. Still, the upper bounds are problem-specific and depend on the density estimator used, the nature of the data, and the dataset size. In this work, we analyzed MMLU, which has a semantically rich set of topics characterized by good enough sampling. In the Appendix, in Sec. D, we show that our results extend to another dataset mixture with a subject partition similar to MMLU. However, the analysis of other QA datasets and generic textual sources would make our observations more general. The prompt structure can also be made more general. In the current study, we studied ICL framed as few-shot learning but further investigations on more differentiated contexts would strengthen our findings. Finally, the description of the transition observed in the proximity of half of the network can be analyzed more in detail, for instance, by providing an interpretation of the *mechanism* [63] underlying the information flow from the context to the last token position [64].

## Acknowledgements and disclosure of funding

We thank Alessandro Laio for many helpful discussions and valuable suggestions. We also thank the technical support of the Laboratory of Data Engineering staff and acknowledge the AREA Science Park supercomputing platform ORFEO.

A.A., A.C., and D. D. were supported by the project "Supporto alla diagnosi di malattie rare tramite l'intelligenza artificiale"- CUP: F53C22001770002. A.A., A. C. were supported by the European Union – NextGenerationEU within the project PNRR "PRP@CERIC" IR0000028 - Mission 4 Component 2 Investment 3.1 Action 3.1.1. A.S. was supported by the project PON "BIO Open Lab (BOL) - Raforzamento del capitale umano"- CUP: J72F20000940007.

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

# Appendix

## A  Fine-tuning setup.

The dataset on which we fine-tune the models is the union of the MMLU dev set (all five examples per subject are selected) and a subset of the MMLU validation set. We select up to 20 examples per subject for Llama2-7b and Llama2-13b and up to 40 examples per subject for the rest of the models. In the first case, the dataset size is 1065, and the second is 1439. We train Llama3-8b and Mistral-7b for 6 epochs and the remaining models for 4.

We fine-tune the models with LoRA. The LoRA rank is 64, $\alpha$ is 16, and dropout = 0.1. For the 70 billion models, we choose a rank of 128, and $\alpha$ is 32. This is an empirically reasonable choice since the embedding dimension of these models is two times larger than the 7 billion ones. We use a learning rate = $2 \cdot 10^{-4}$; for the 70 billion models we decrease it to $1 \cdot 10^{-4}$. For all the models, we apply a cosine annealing scheduler and a linear warm-up for 5% of the total iterations. We fine-tune all the models with batch size = 16 using the Adam optimizer without weight decay.

Below, we report the performance of the MMLU test set on all the 14042 samples, for 0-shot, 5-shot, and fine-tuned models. To compute the *macro* average, we first measure the accuracy for each subject. Then, we compute the arithmetic mean of the subject accuracies. The *micro* average is the fraction of correct answers taken over the dataset. The two quantities differ since the dataset is unbalanced.

Table A1: Zero and 5-shot accuracies. We report micro and macro averages on the MMLU test set.

| model | num shots | accuracy % (macro/micro) |
|---|---|---|
| Llama3-8b | 0 shot | 63.7 / 62.1 |
| Llama3-8b | 5 shot | 66.5 / 65.1 |
| Llama3-70b | 0 shot | 76.7 / 75.0 |
| Llama3-70b | 5 shot | 79.2 / 78.5 |
| Mistral-7b | 0 shot | 57.8 / 55.9 |
| Mistral-7b | 5 shot | 63.7 / 62.1 |
| Llama2-7b | 0 shot | 38.9 / 37.5 |
| Llama2-7b | 5 shot | 46.6 / 45.9 |
| Llama2-13b | 0 shot | 52.9 / 52.0 |
| Llama2-13b | 5 shot | 55.4 / 54.8 |
| Llama2-70b | 0 shot | 66.3 / 65.5 |
| Llama2-70b | 5 shot | 69.5 / 68.8 |

Table A2: Fine-tuning accuracies. We report the micro and macro averages on the MMLU test set. We only report the micro average on the train sets.

| model | epochs | accuracy % test (macro/micro) | accuracy % train (micro) |
|---|---|---|---|
| Llama3-8b | 6 | 65.6 / 64.8 | 94.8 |
| Llama3-70b | 4 | 78.5 / 79.0 | 93.8 |
| Mistral-7b | 6 | 61.4 / 59.9 | 96.4 |
| Llama2-7b | 4 | 51.9 / 50.7 | 73.8 |
| Llama2-13b | 4 | 56.1 / 55.5 | 79.7 |
| Llama2-70b | 4 | 69.9 / 71.1 | 92.4 |

## B    Iterative search of density peaks and saddle points.

Let $\mathcal{N}_k(i)$ be the set of $k$ points nearest to $\mathbf{x}_i$ in Euclidean distance at a given layer $l$.

The first step of the density-peaks clustering is finding the point of maximum density $\rho_i$ (namely the probability peaks). Data point $i$ is a maximum if the following two properties hold: (I) $\rho_i > \rho_j$ for all the points $j$ belonging to $\mathcal{N}_k(i)$; (II) $i$ does not belong to the neighborhood $\mathcal{N}_k(j)$ of any other point of higher density [11]. (I) and (II) must be jointly verified as the neighborhood ranks are not symmetric between pairs of points. A different integer label $\mathcal{C} = \{c^1, ...c^n\}$ is then assigned to each of the $n$ maxima. The data points that are not maxima are iteratively linked to one of these labels by assigning the same label as its nearest neighbor of higher density to each point. The set of points with the same label corresponds to a cluster.

The saddle points between two clusters are identified as the points of maximum density between those lying on the borders between the clusters. A point $\mathbf{x}_i \in c^\alpha$ is assumed to belong to the border $\partial_{c^\alpha, c^\beta}$ with a different peak $c^\beta$ if exists a point $\mathbf{x}_j \in \mathcal{N}_k(i) \cap c^\beta$ whose distance from $i$ is smaller than the distance from any other point belonging to $c^\alpha$. The saddle point between $c^\alpha$ and $c^\beta$ is the point of maximum density in $\partial_{c^\alpha, c^\beta}$.

## C    The Adjusted Rand Index.

To determine to which degree the density peaks are consistent with the abstractions of the data, we will compare the clustering induced by the density peaks algorithm with the partition given by the 300 classes and that given by the high-level subdivision in animals and objects. Among the many possible scores, we choose the Adjusted Rand Index (ARI) [54], one of the best clusters of external evaluation measures according to [65]. The Rand Index (RI) [66] measures the consistency between a cluster partition $\mathcal{C}$ to a reference partition $\mathcal{R}$ counting how many times a pair of points:

$a$ are placed in the same group in $\mathcal{C}$ and $\mathcal{R}$;

$b$ are placed in different groups in $\mathcal{C}$ and $\mathcal{R}$;

$c$ are placed in different groups in $\mathcal{C}$ but in the same group in $\mathcal{R}$;

$d$ are placed in the same group in $\mathcal{C}$ but in different groups in $\mathcal{R}$;
and measures the consistency with $RI = (a+b)/(a+b+c+d)$. The Rand Index is not corrected for chance, meaning it does not give a constant value (e.g., zero) when the two assignments are random. [54] proposed to adjust $RI$, taking into account the expected value of the Rand Index, $n_c$, under a suitable null model for chance:

$$ARI = \frac{a + b - n_c}{a + b + c + d - n_c} \tag{1}$$

$ARI$ is equal to 1 when the two partitions are consistent and 0 when the assignments are random and can take negative values. A large value of $ARI$ not only implies that instances of the same class are put in the same cluster (homogeneity) but also that the data points of a class are assigned to a single cluster (completeness).

# D Additional experiments

## D.1 Analysis on an additional dataset mixture.

In this section, we validate the findings shown in Section 4.1 on a dataset constructed from TheoremQA [67], ScibenchQA [68], Stemez [3], and RACE [69]. This dataset contains roughly 6700 examples not included in MMLU, ten subjects from STEM topics, and a middle school reading comprehension task (RACE), with at least 300 examples per subject. We keep four choices for each answer. The 0-shot, 5-shot, and fine-tuned accuracies in Llama3-8b are 55%, 57%, and 58%, respectively.

In Fig. A1-left, we see that the intrinsic dimension profiles have a peak around layers 15/16, the *same layers as in MMLU* (see Fig. 2-left). This peak in ID signals the transition between the two phases described in the paper. Before layer 17, few-shot models encode better information about the subjects (ARI with subjects above 0.8). Between layers 3 and 7, the peaks in 5-shot layers reflect the semantic similarity of the subjects (see the dendrograms for layer five reported in Fig. A2).

Fine-tuning instead changes the representations after layer 17, where ARI with the answers for the is ∼ 0.15, higher than that of the 5-shot and 0-shot models. The absolute value is lower than that reported in the main paper (Fig. 5-left) because the fine-tuned accuracy reached on the STEM subjects in this dataset is lower. Overall, the results are consistent with those shown in the paper for MMLU.

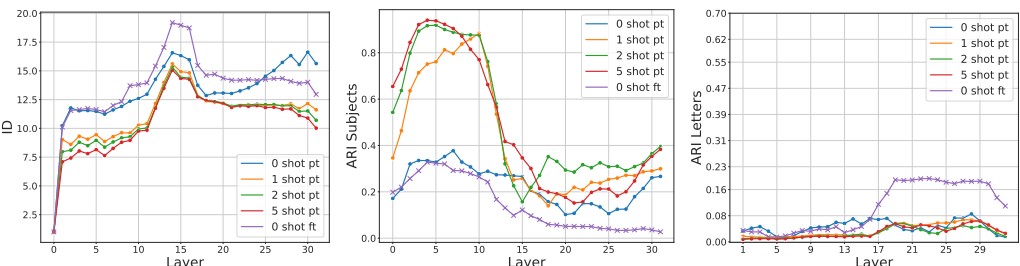

Figure A1: **Intrinsic dimension (left), ARI with the subjects (center) and with the answers (right) in Llama 3 8b.** The dataset is constructed from *Scibench*, which has 541 QA pairs about chemistry, math, physics), *TheoremQA* with 598 QA pairs about business, computer science, math, and physics, *Stemez* with 4083 QA pairs about biology, business, chemistry, computer science, economics, engineering, physics, psychology, and the test set of *RACE* with 1436 QA pairs about middle school tests. We finetuned Llama 3 8b, keeping 50 examples per topic.

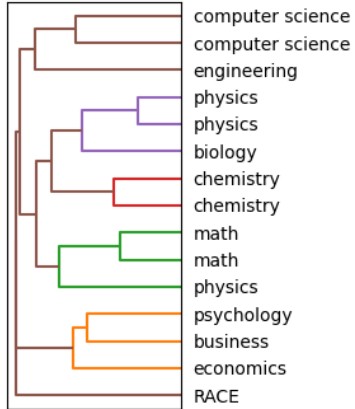

Figure A2: **Organization of the density peaks of layer 5 of Llama 3 8b in the 5-shot setup.**

## D.2  Intrinsic dimension profiles.

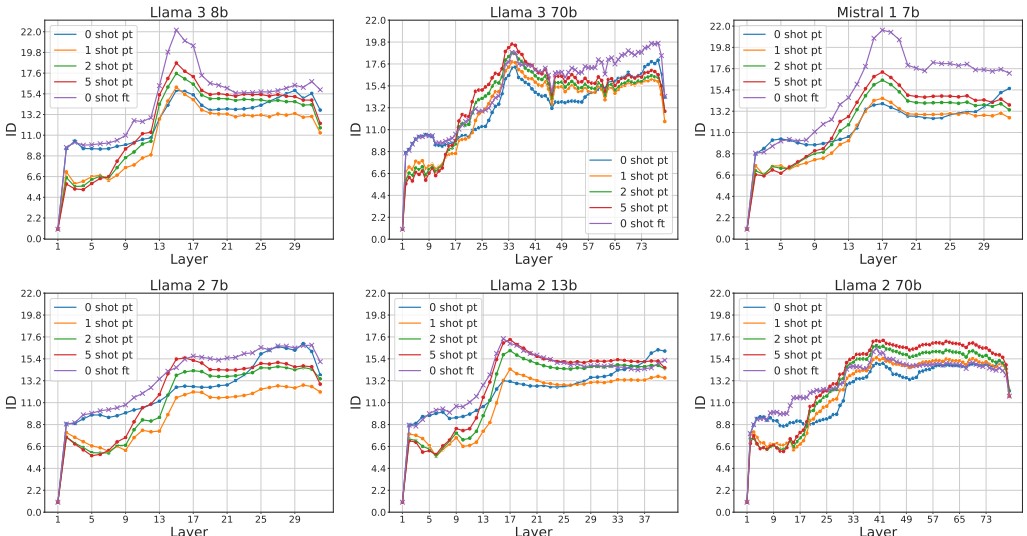

Figure A3: **Intrinsic dimension (ID) of the data representations in hidden layers of large transformers** for increasing number of shots in the prompt and fine-tuned representations. Llama 3 8b (upper left), Llama 3 70b (upper center), Mistral 7b (upper right), Llama 2 7b (lower left), Llama 2 13b (lower center), and Llama 2 70b (lower right). For each layer, we calculated the intrinsic dimension of the hidden representations produced at the last token of the prompt, which provides the answer. The algorithm used is Gride with $k = 16$ nearest neighbors considered. We observe that all models exhibit similar behavior in the early layers regardless of the number of shots in the prompt. However, from the middle of the network onwards, the profiles diverge. Configurations that achieve a higher agreement between clusters and the letter given as the answer (as shown in A10) in the last layers will have a lower intrinsic dimension. These results support the intuitive observation that fewer features are needed to represent the information as the model starts to encode the letter it will provide as the answer.

### D.2.1  Scale analysis of the intrinsic dimension.

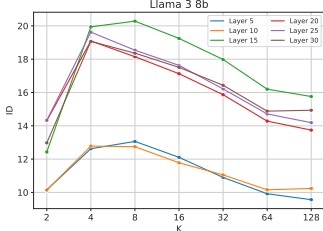

Figure A4: **Scale analysis of the intrinsic dimension for Llama 3 8b**. The plot presented herein depicts the intrinsic dimension (ID) at layers 5, 10, 15, 20, 25, and 30, varying the number of nearest neighbors $k$ utilized by the Gride algorithm. Following the methodology delineated in [50, 51]; we observe the evolution of the ID as a function of $k$ to identify a plateau in the estimates. After initial growth, from $k = 8$ to $k = 16$, the ID stabilizes and then decreases. Consequently, we select $k = 16$ as it represents the scale appropriate for examining the topology of the data being.

## D.3 Number of Clusters

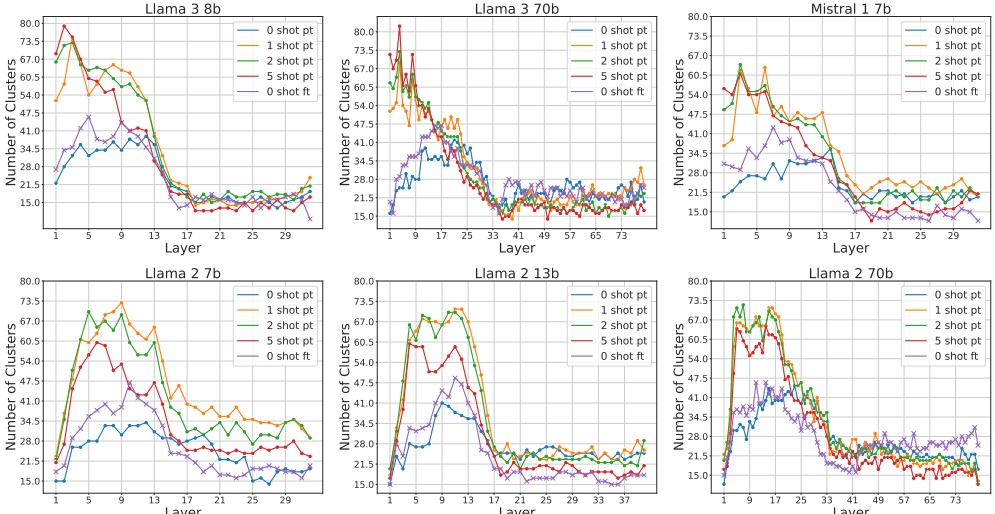

Figure A5: **Number of clusters across layers in large transformers**. The plots above show the number of clusters identified by the Adaptive Density Peak (ADP) clustering algorithm at each model layer. In configurations where multiple examples are provided in the context, we can observe that the number of clusters closely matches the 57 subject categories present in the MMLU dataset. This phenomenon aligns with the observation that the agreement between the cluster partition and the subject-induced partition from MMLU is primarily influenced by the number of examples provided, as illustrated in Figure 3.

## D.4 Core points

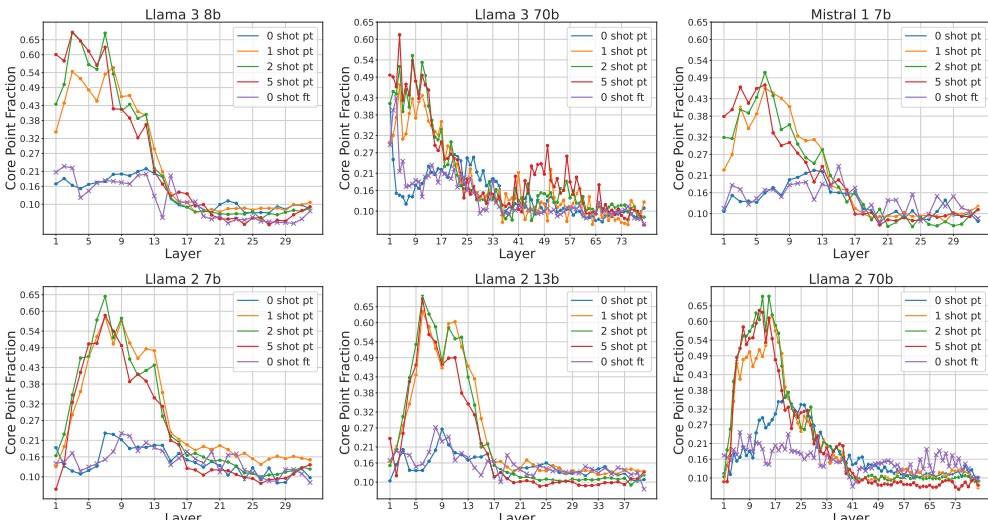

Figure A6: **Number of core points across layers in large transformers**. The figure shows the fraction of points classified by ADP as "core," indicating they were assigned to a cluster with higher confidence. Conversely, the algorithm designates points with lower densities than the highest border density between clusters as "halo" and thus, these points are discarded.

## D.5 Experiments on Llama2-7b, Llama2-13b and Llama2-70b.

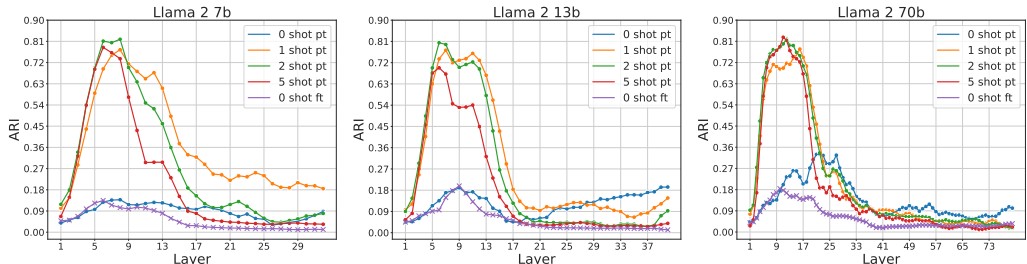

Figure A7: **Adjusted Rand Index (ARI) between clusters and MMLU subjects (test set)**. The figure shows the ARI with the subjects for Llama2-7b (left), Llama2-13b (center), and Llama2-70b (right) for increasing the number of shots in the prompt and fine-tuned representations. The plot illustrates the agreement between the partition generated by the clusters and the partition induced by the subject labels from the MMLU dataset. The clustering was performed using Density Peak Clustering with Z = 1.6, the default value of the DADAPy package. As depicted in 3, the agreement is higher at the initial layers of the network and increases with a growing number of examples provided in the context. In other words, the clustering aligns more closely with the subject label in the early layers of the network in the few-shot setting.

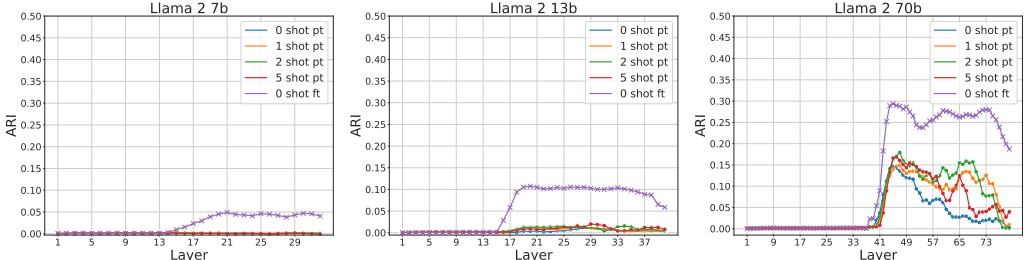

Figure A8: **Adjusted Rand Index between clusters and final answers.** Llama2-7b (left), Llama2-13b (center), and llama2-70b (right) for increasing number of shots in the prompt and fine-tuned representations. The plot depicts the correspondence between the partitions generated by the clusters and the partitions induced by the model's predicted answer letters. The clustering was performed using Density Peak Clustering with Z = 1.6, the default value of the DADAPy package. As shown in 3, we can observe a trend that is opposite to the one seen with the subject label: the purity of the clusters with respect to the answer partition is higher in the final layers of the model, and it does not depend on the number of examples provided in the prompt. In this case, the fine-tuned model with zero examples in the context (0-shot) achieves the highest Adjusted Rand Index (ARI), indicating the highest agreement between the clustering and the model's predicted answer partitions.

## D.6 Profiles without moving average

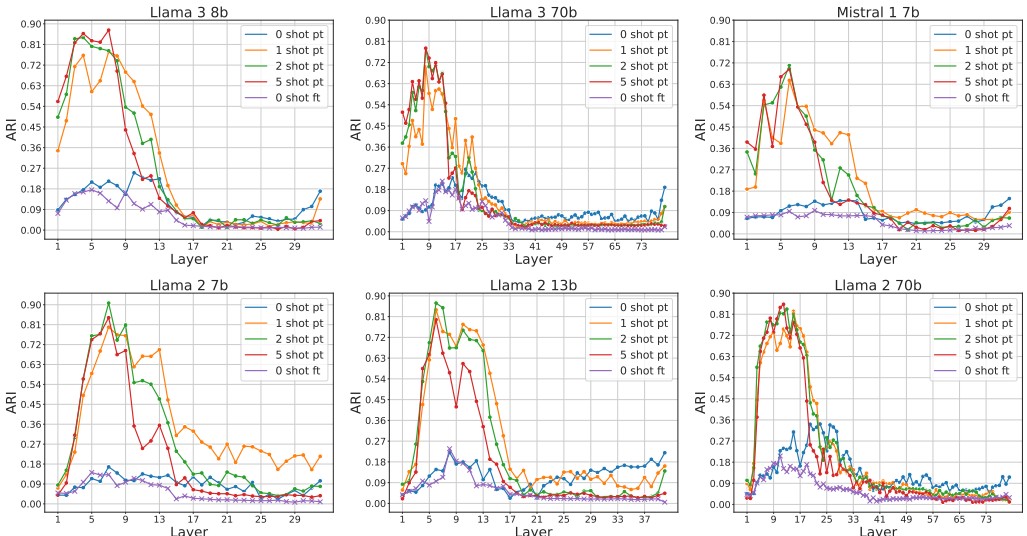

Figure A9: **Adjusted Rand Index (ARI) between clusters and MMLU subjects (test set) without moving average**. Llama 3 8b (upper left), Llama 3 70b (upper center), Mistral 7b (upper right), Llama 2 7b (lower left), Llama 2 13b (lower center), and Llama 2 70b (lower right). The plots are produced on the same data as the one in A7, but in this case, we displayed the result without averaging over range of successive layers

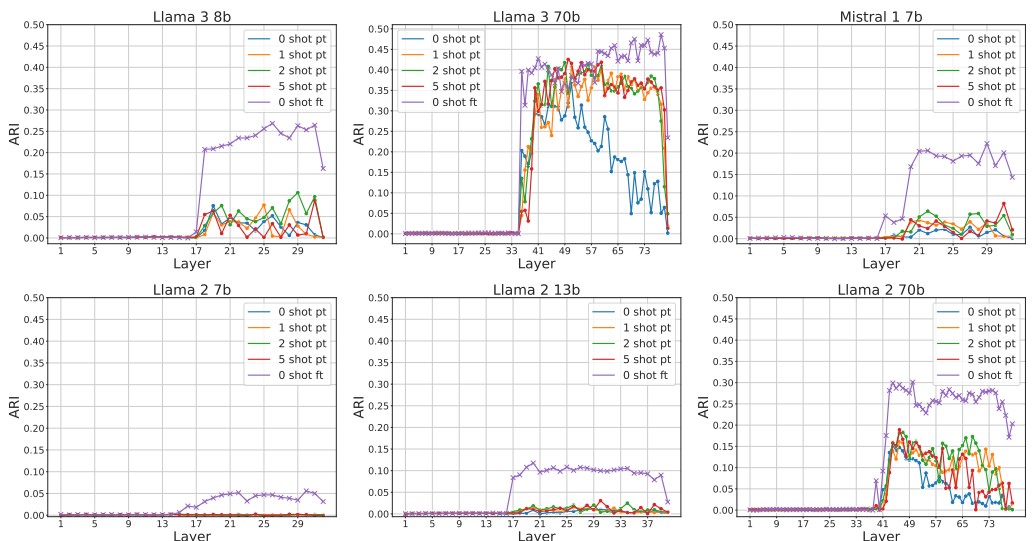

Figure A10: **Adjusted Rand Index between clusters and final answers without moving average.** Llama 3 8b (upper left), Llama 3 70b (upper center), Mistral 7b (upper right), Llama 2 7b (lower left), Llama 2 13b (lower center), and Llama 2 70b (lower right). The plots are produced on the same data as the one in 5, but in this case, we displayed the result without averaging over a range of successive layers

## D.7 Dendrograms (5-shot) in layers where the subject ARI is highest

### D.7.1 Llama3-8b

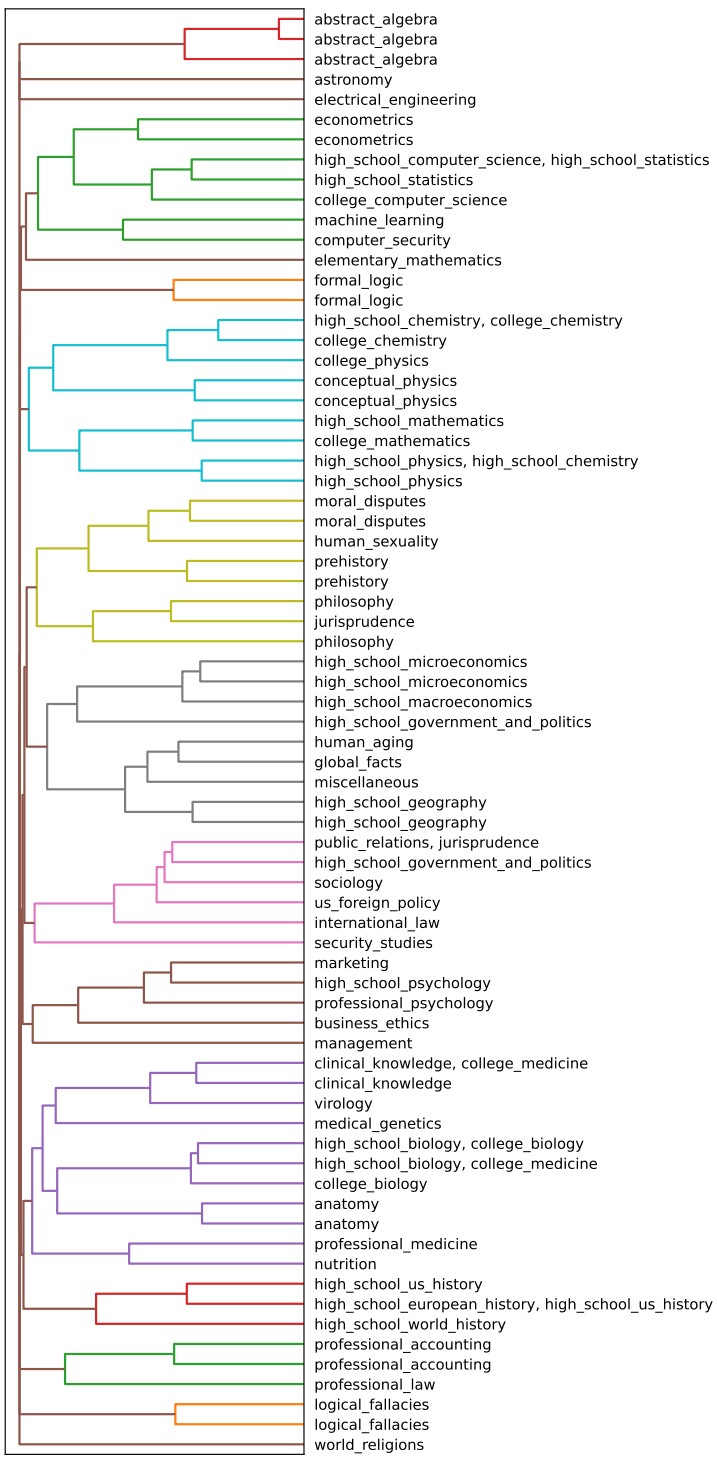

Figure A11: **Llama3-8b, 5-shot, Z=1.6, ARI = 0.82, 77 clusters**

### D.7.2 Llama3-70b

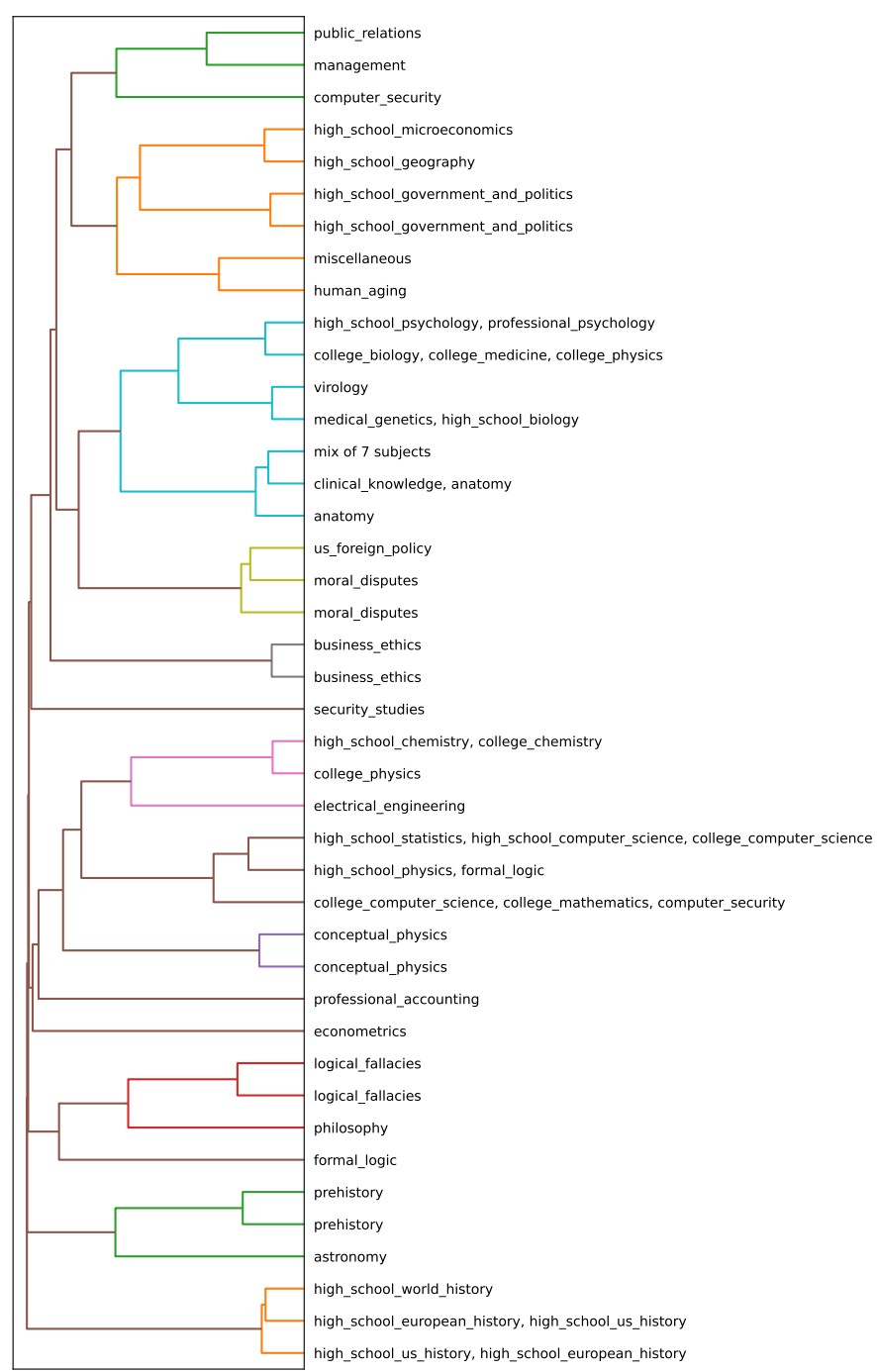

Figure A12: **Llama3-70b, 5-shot, Z=1.6, ARI = 0.65, 69 clusters**

### D.7.3 Mistral-7b

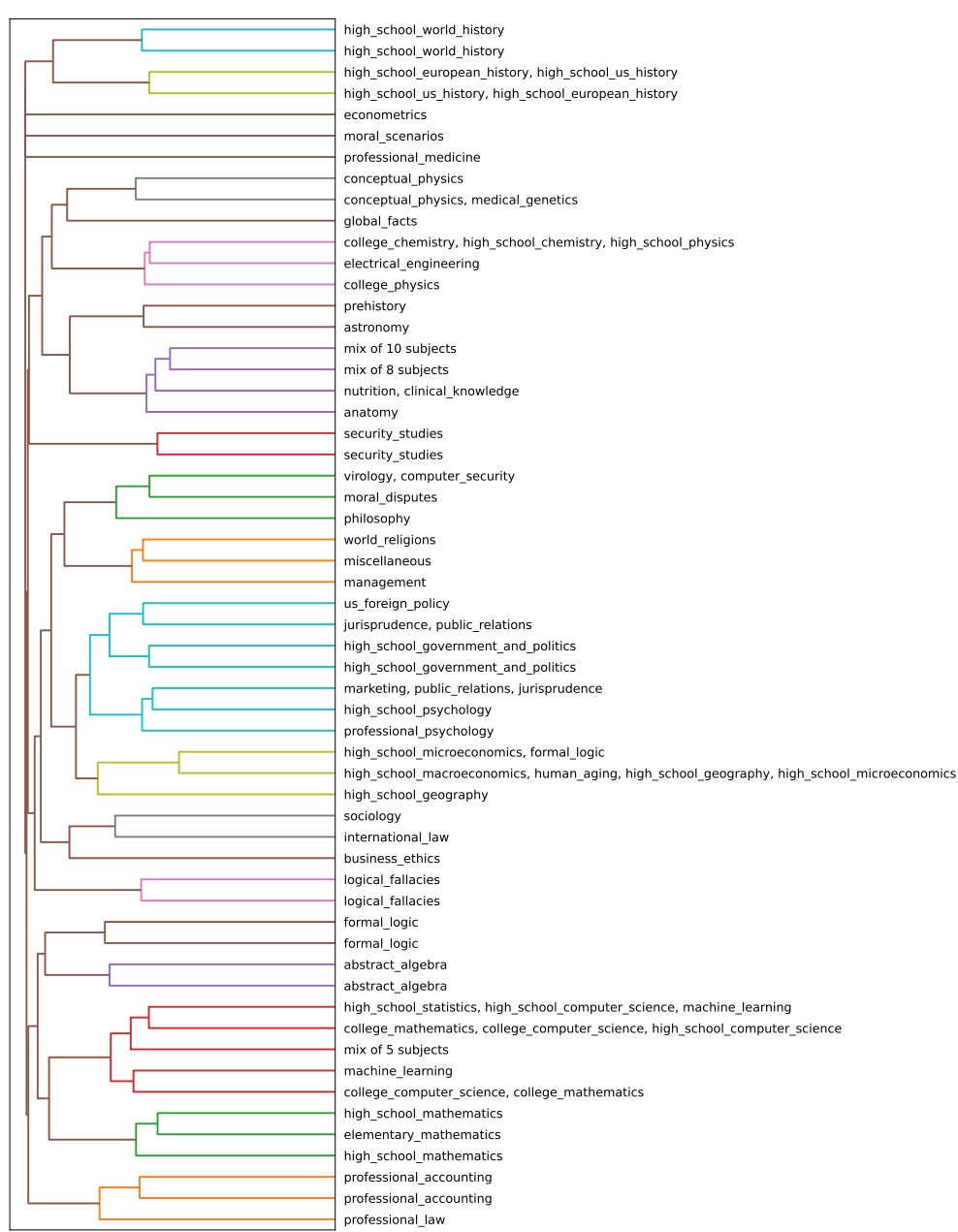

Figure A13: **Mistral, 5-shot, Z=1.6, ARI = 0.53, 57 clusters**

### D.7.4 Llama2-7b

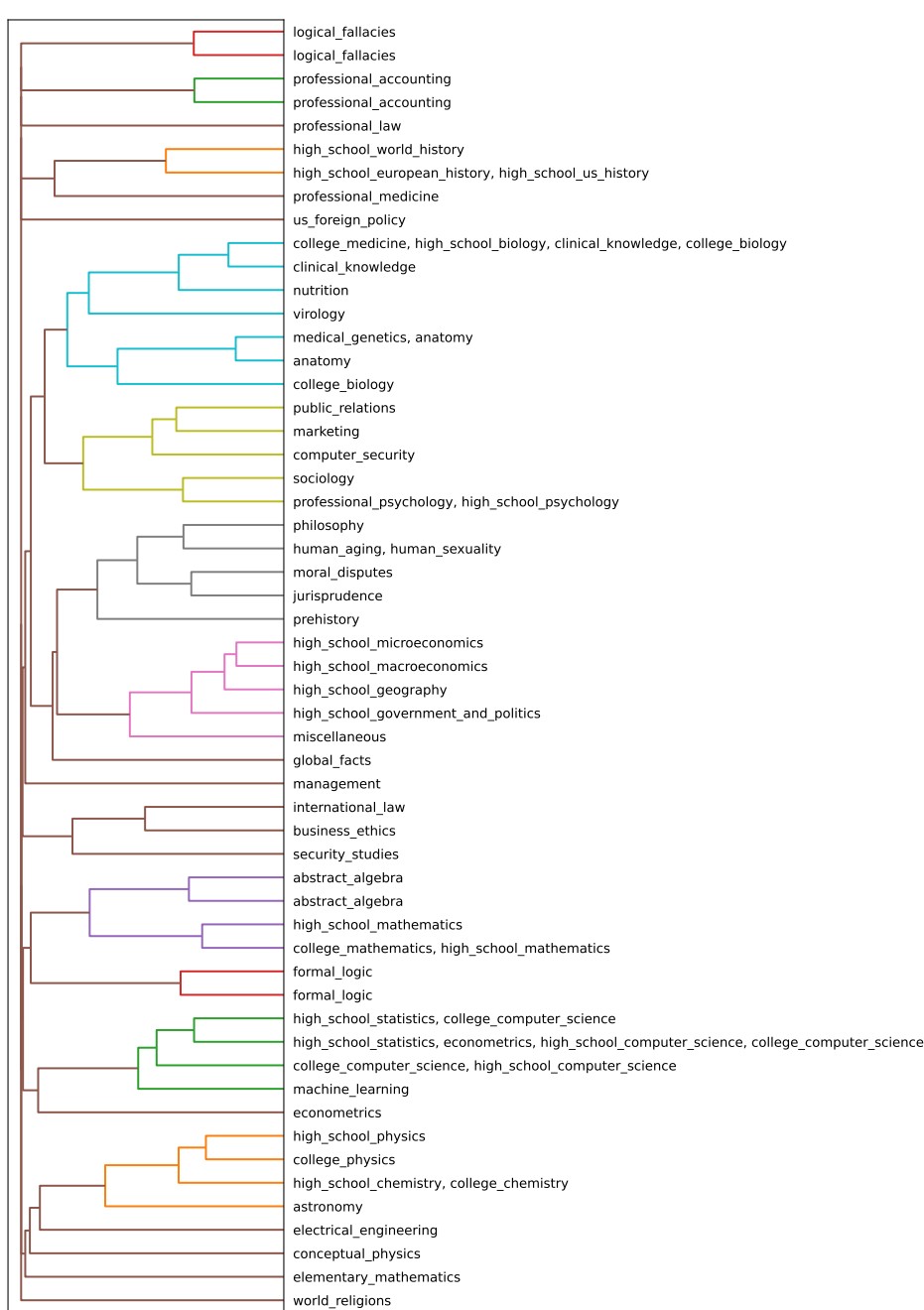

Figure A14: **Llama2-7b, 5-shot, Z=1.6, ARI = 0.76, 60 clusters**

### D.7.5 Llama2-13b

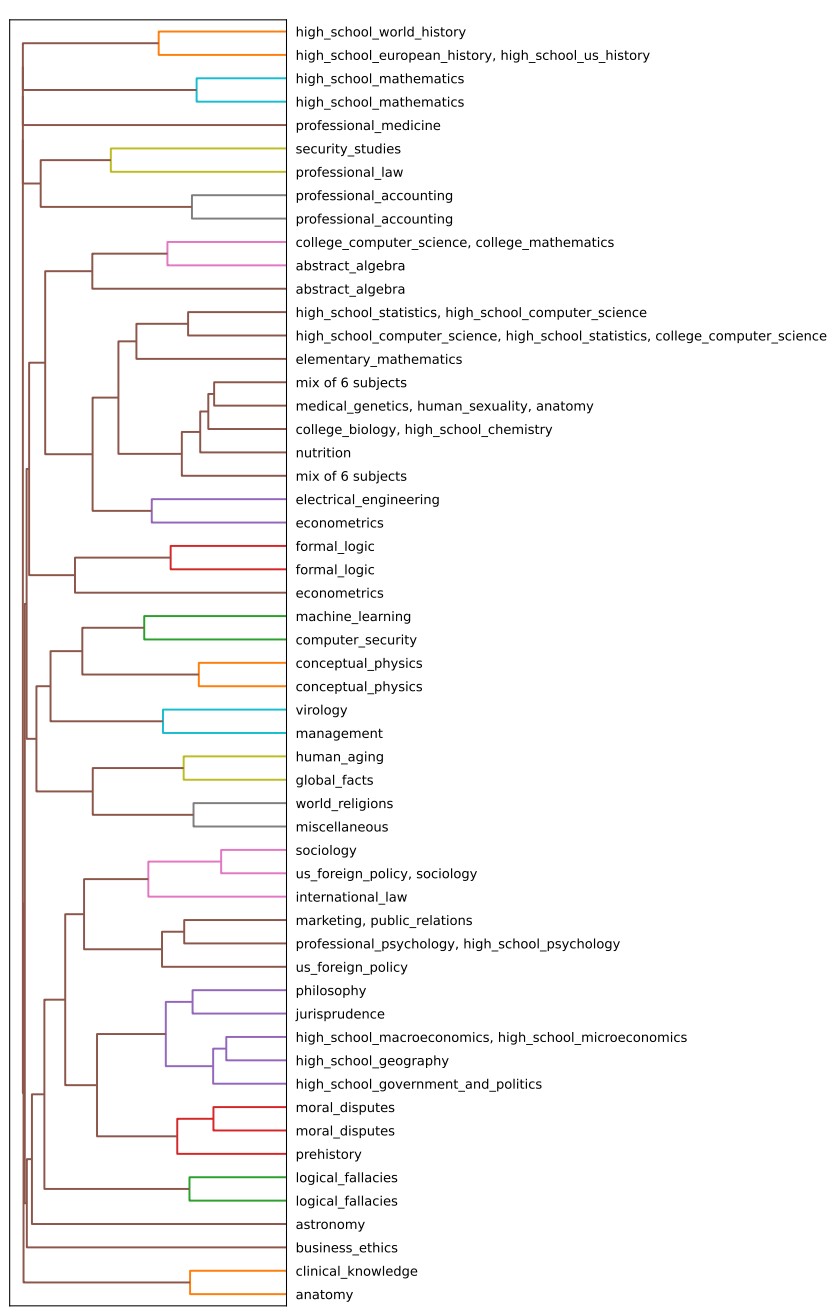

Figure A15: **Llama2-13b, 5-shot, Z=1.6, ARI = 0.64, 59 clusters**

### D.7.6 Llama2-70b

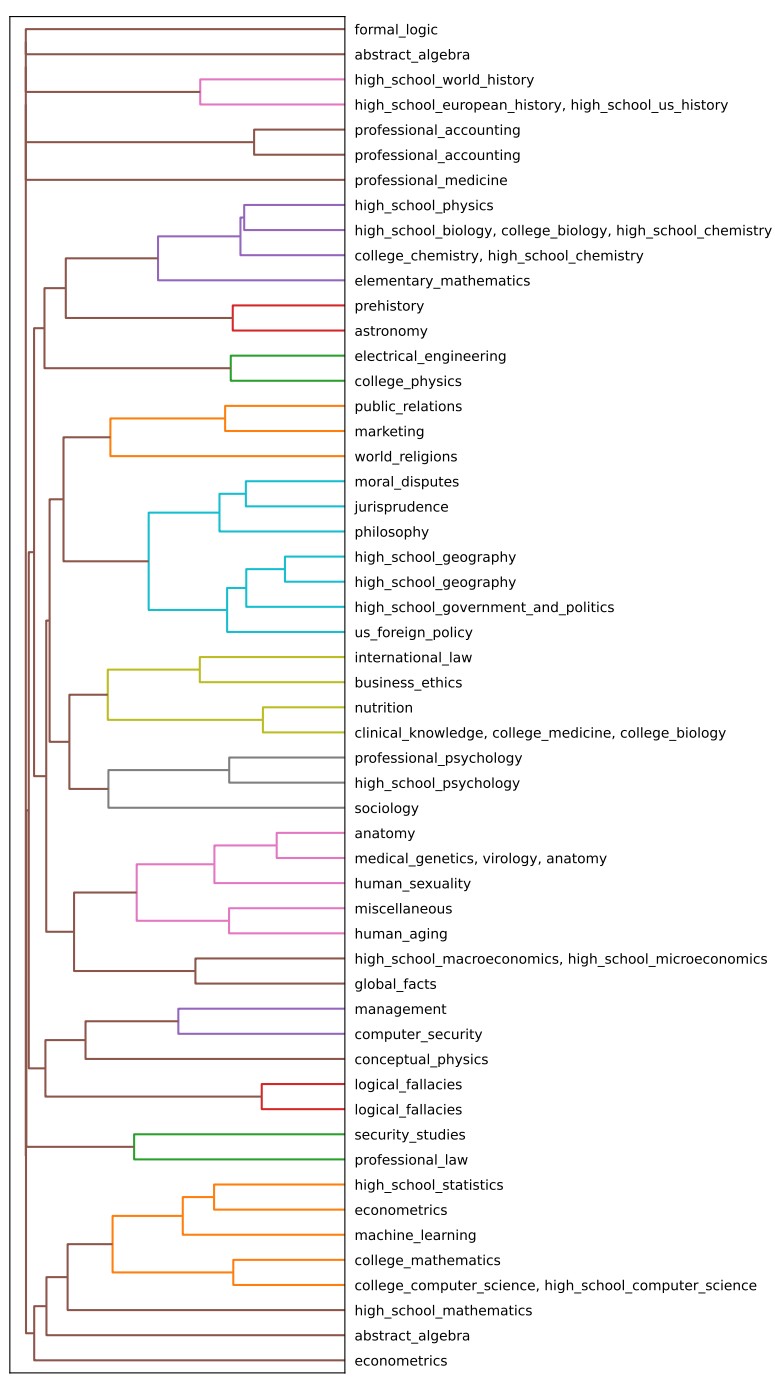

Figure A16: **Llama2-70b, 5-shot, Z=1.6, ARI = 0.80, 56 clusters**

# E  Robustness with respect to Z

## E.1  ARI profiles with letters

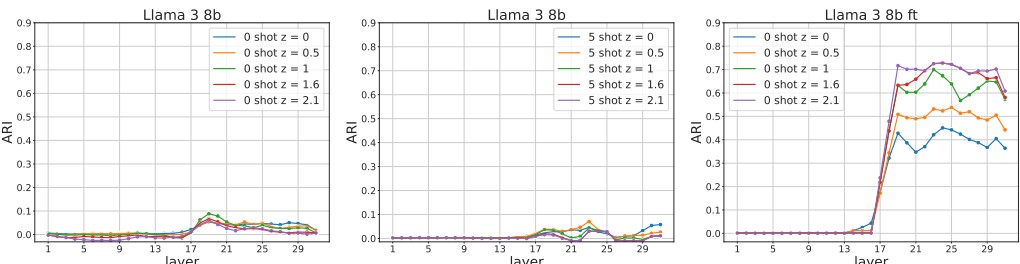

Figure A17: **The Adjusted Rand Index (ARI) between clusters and the MMLU answer letters (test set) varying free parameter** $Z$**.** Llama 3 8b (left, center) and Llama 3 8b fine-tuned (right). We replicated the experiment depicted in 5, monitoring the changes in ARI as a function of $Z$. The results indicate that the metric is robust to small variations in the free parameter, with all three configurations showing that a change of $Z = 2.1$ corresponds to an approximate change of $0.3$ in the resulting ARI.

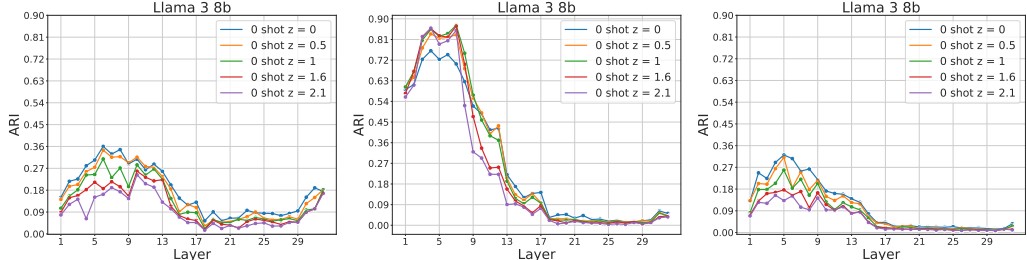

Figure A18: **The Adjusted Rand Index (ARI) between clusters and the MMLU subjects (test set) varying free parameter** $Z$**.** Llama 3 8b (left, center) and Llama 3 8b fine-tuned (right). We replicated the experiment depicted in 5, monitoring the changes in ARI as a function of $Z$.

## E.2 Dendrograms robustness with respect to Z in Llama3-8b (0-shot, 5-shot, fine-tuned).

The probability landscape of the 0-shot and fine-tuned models remains largely mixed even when $Z$ equals zero, and we consider all the local density maxima as genuine probability modes. For 0-shot, the number of clusters increases from 43 to 70, but we still have two large clusters containing a mixture of 22 and 10 subjects (see Fig. A19). Similarly, the fine-tuned model has several clusters with more than 7 subjects (see Fig. A24). We also notice that the probability landscape of 5-shot representation is more robust to variations of $Z$. Indeed, by increasing $Z$ from 1.6 to 4, the number of density peaks slightly decreases from 77 to 57, and the ARI with the subjects remains around 0.8 (see Fig. A23). In contrast, for the 0-shot and fine-tuned representations, it drops to 0 as most of the probability peaks are merged into large-scale structures (Figs. A21, A26).

### E.2.1 Llama3-8b 0shot Z=0

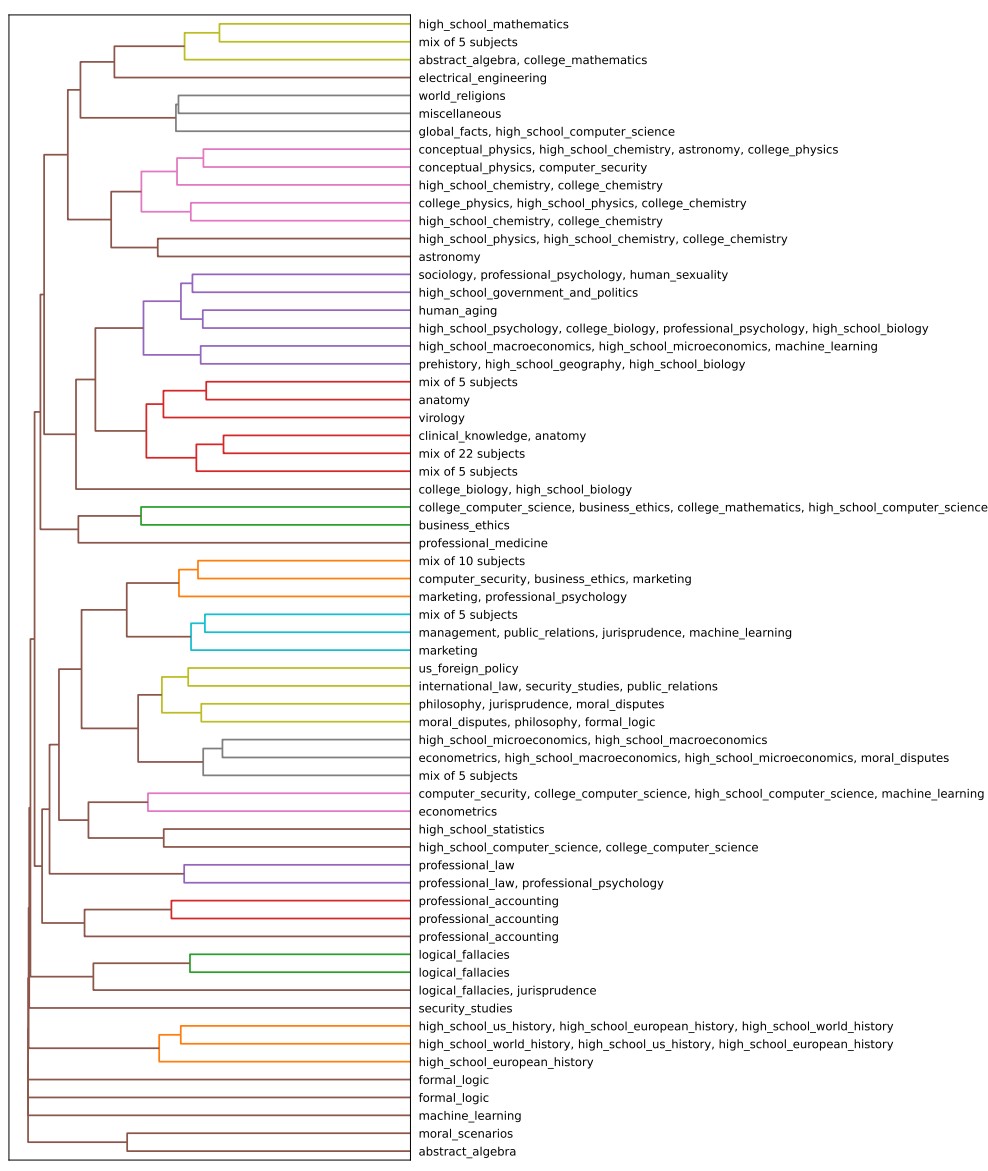

Figure A19: **Llama3-8b, 0-shot, Z=0, ARI = 0.28, 70 clusters**

### E.2.2 Llama3-8b 0shot Z=1.6

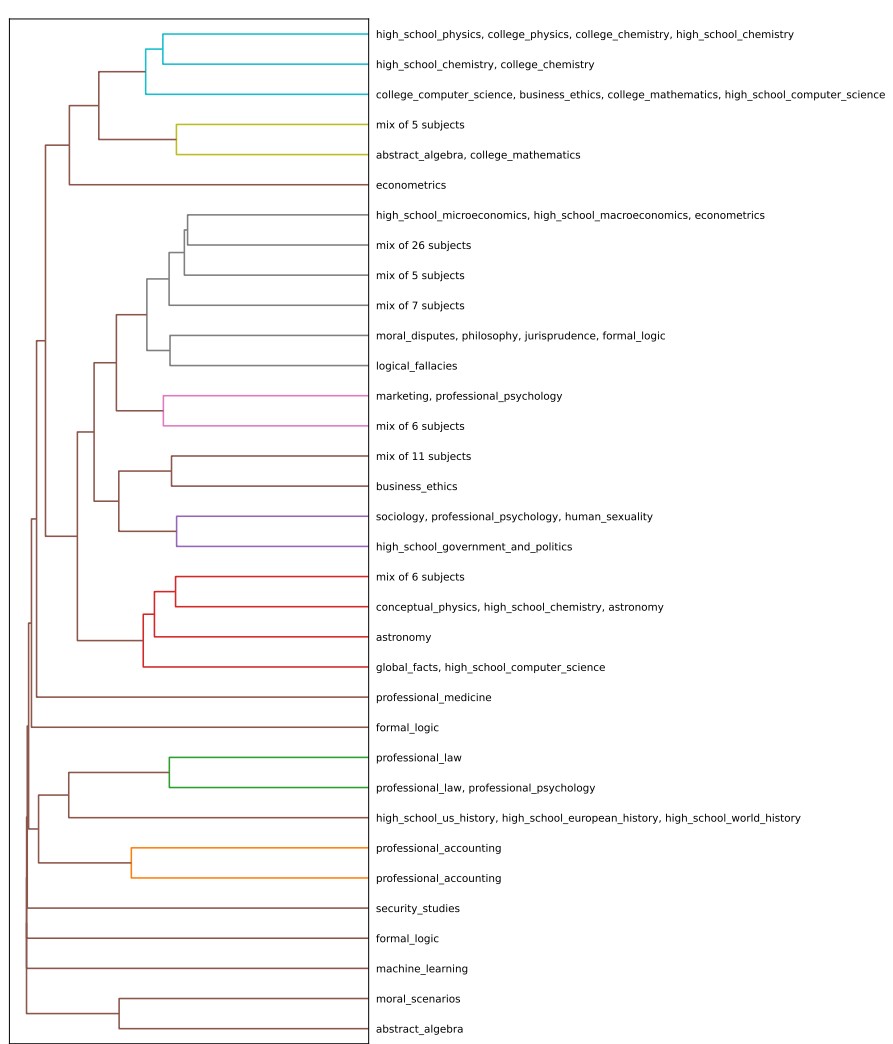

Figure A20: **Llama3-8b, 0-shot, Z=1.6, ARI = 0.15, 34 clusters**

### E.2.3 Llama3-8b 0shot Z=4

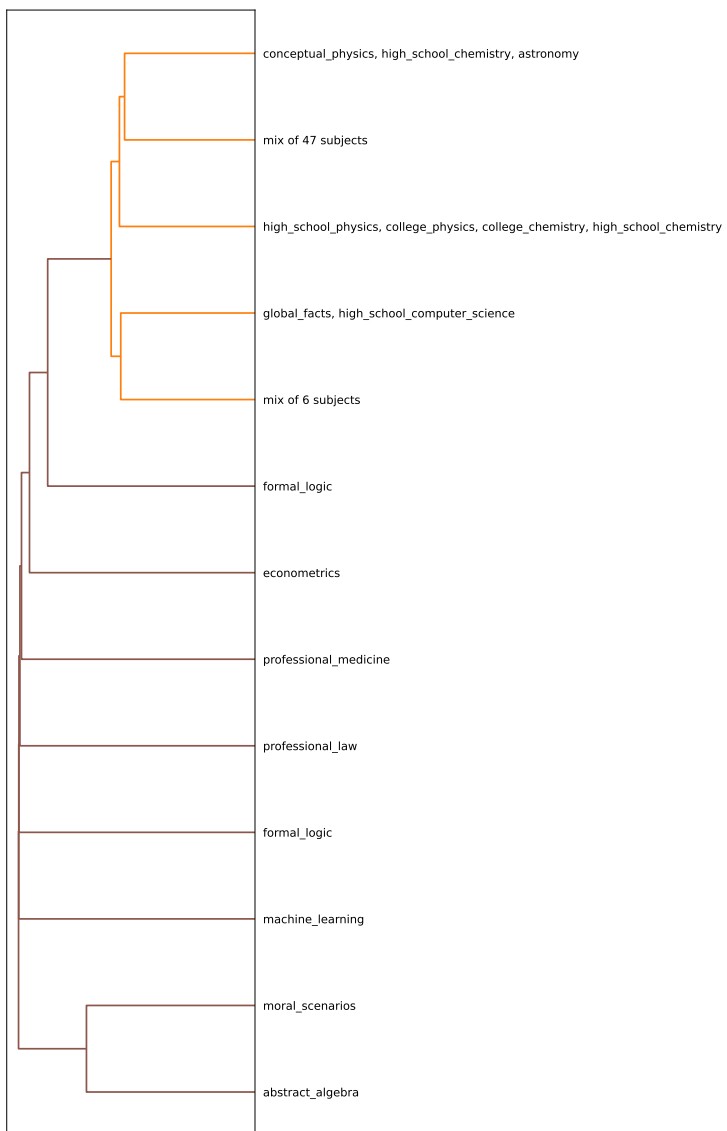

Figure A21: **Llama3-8b, 0-shot, Z=4, ARI = 0.02, 13 clusters**

## E.2.4 Llama3-8b 5shot Z=0

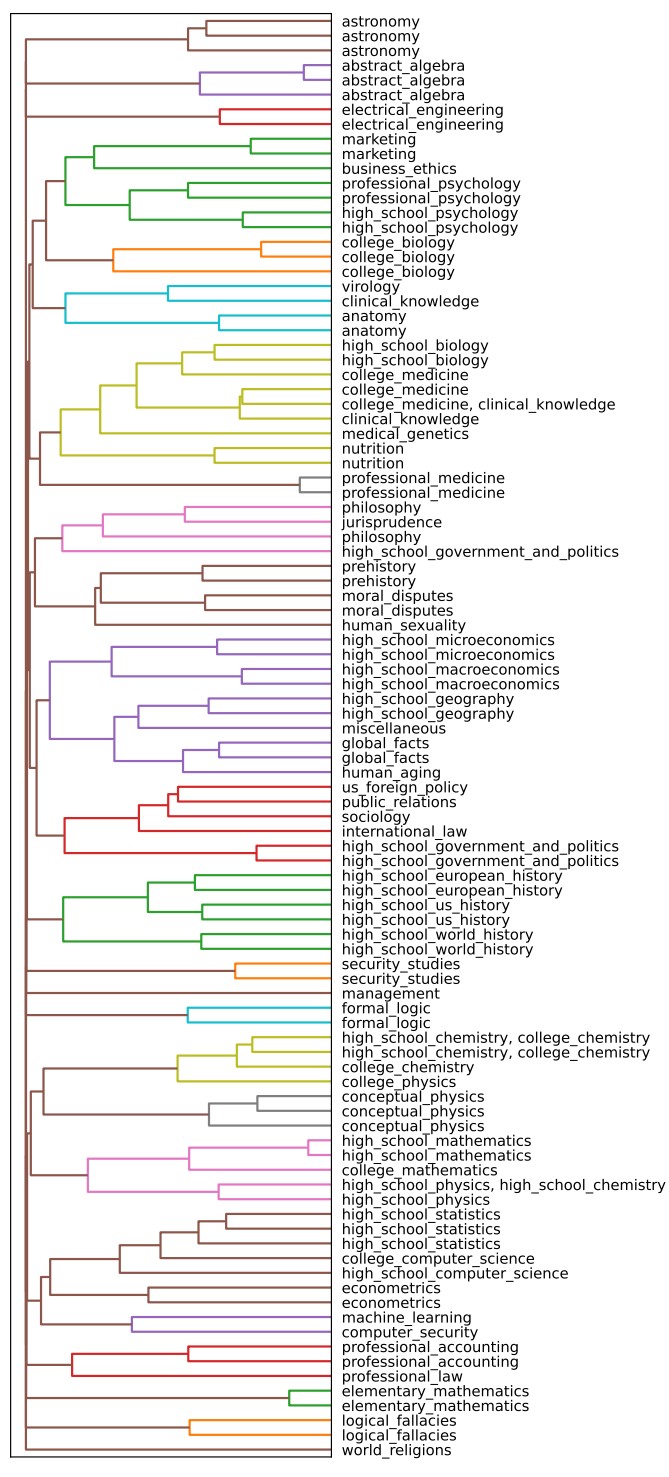

Figure A22: **Llama3-8b, 5-shot, Z=0, ARI = 0.72, 110 clusters**

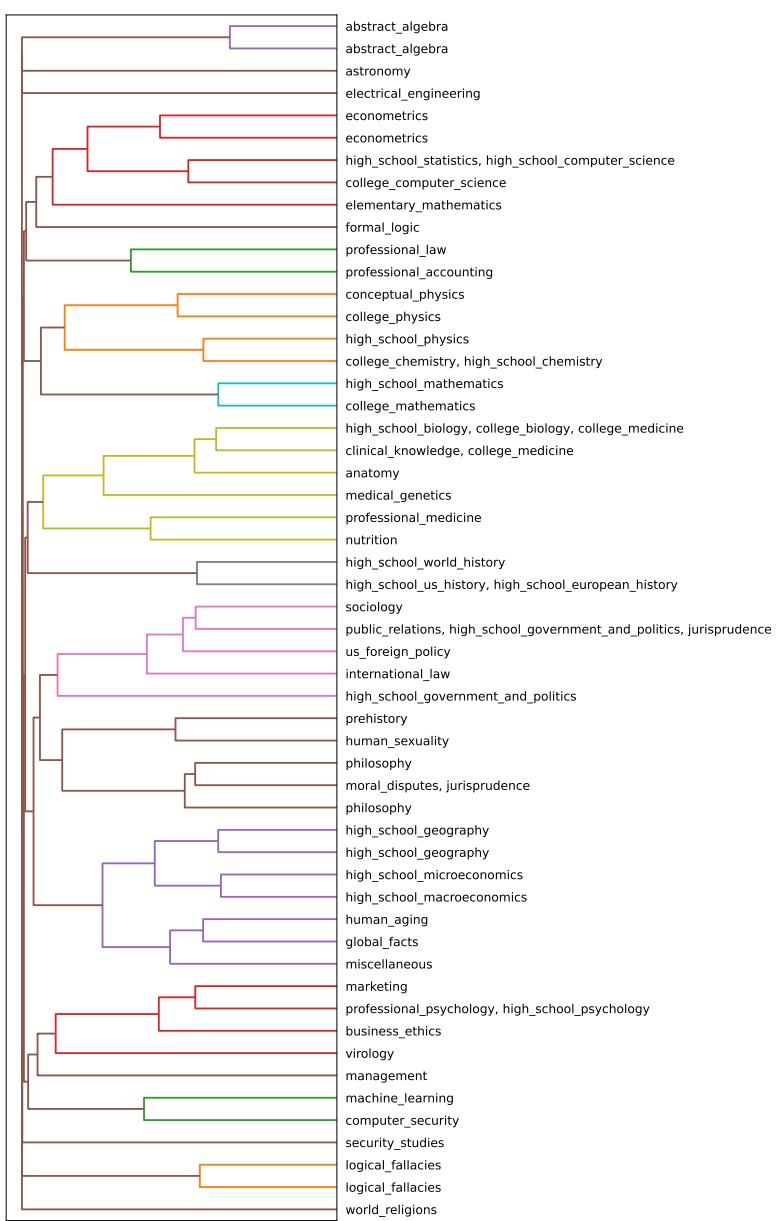

Figure A23: **Llama3-8b, 5-shot, Z=0, ARI = 0.81, 57 clusters**

### E.2.6 Llama3-8b ft Z=0

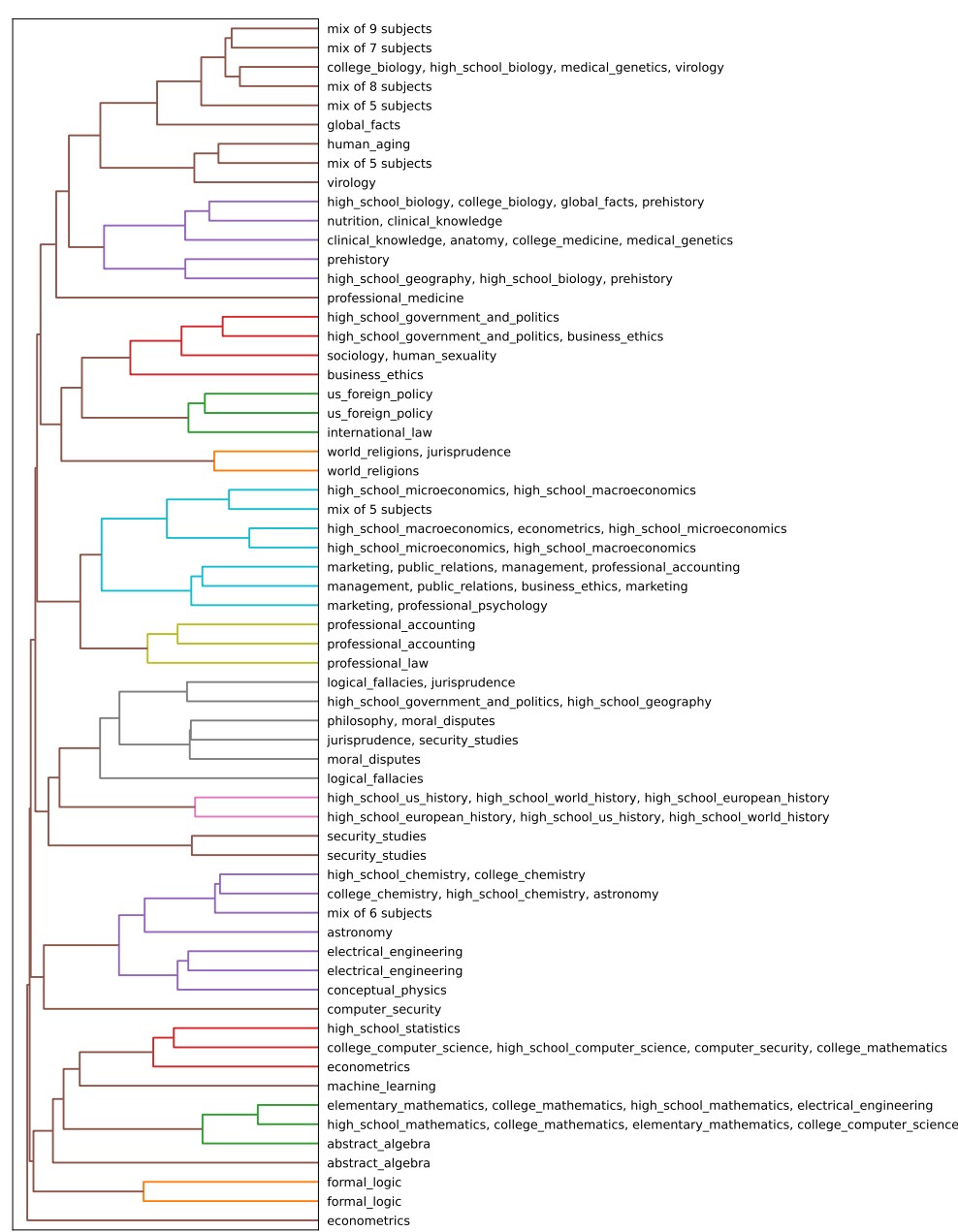

Figure A24: **Llama3-8b, finetuned, Z=0, ARI = 0.35, 69 clusters**

### E.2.7 Llama3-8b ft Z=1.6

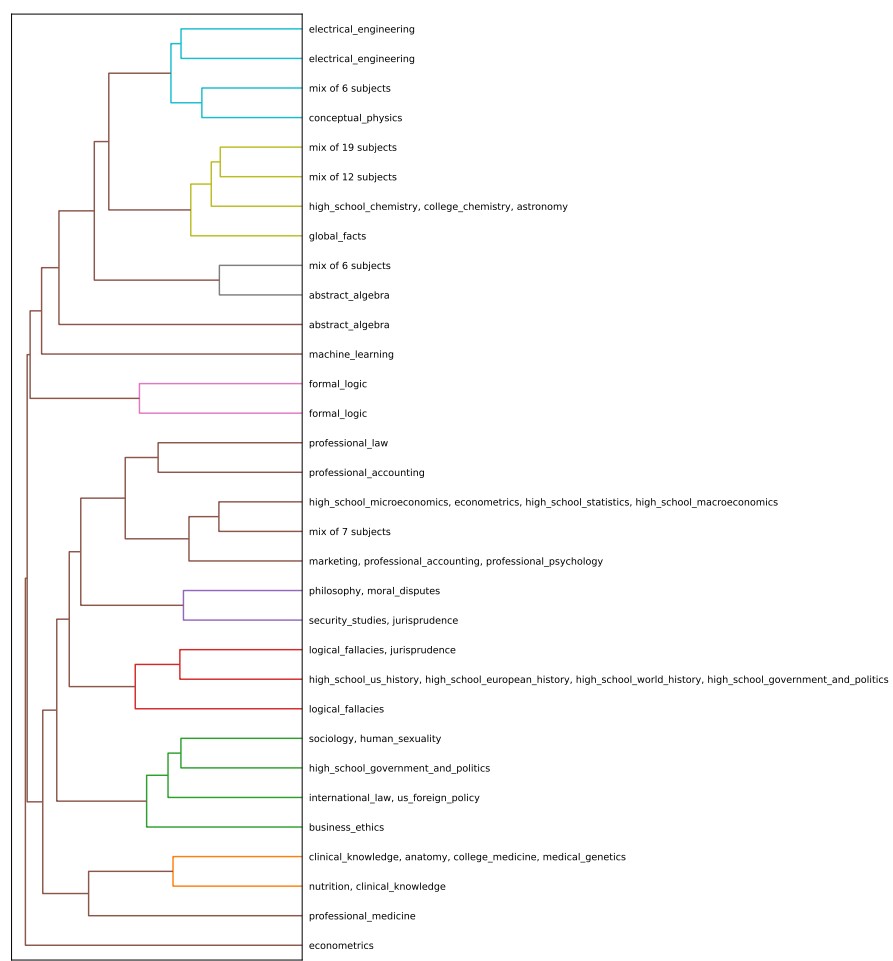

Figure A25: **Llama3-8b, finetuned, Z=1.6, ARI = 0.23, 34 clusters**

### E.2.8 Llama3-8b ft Z=4

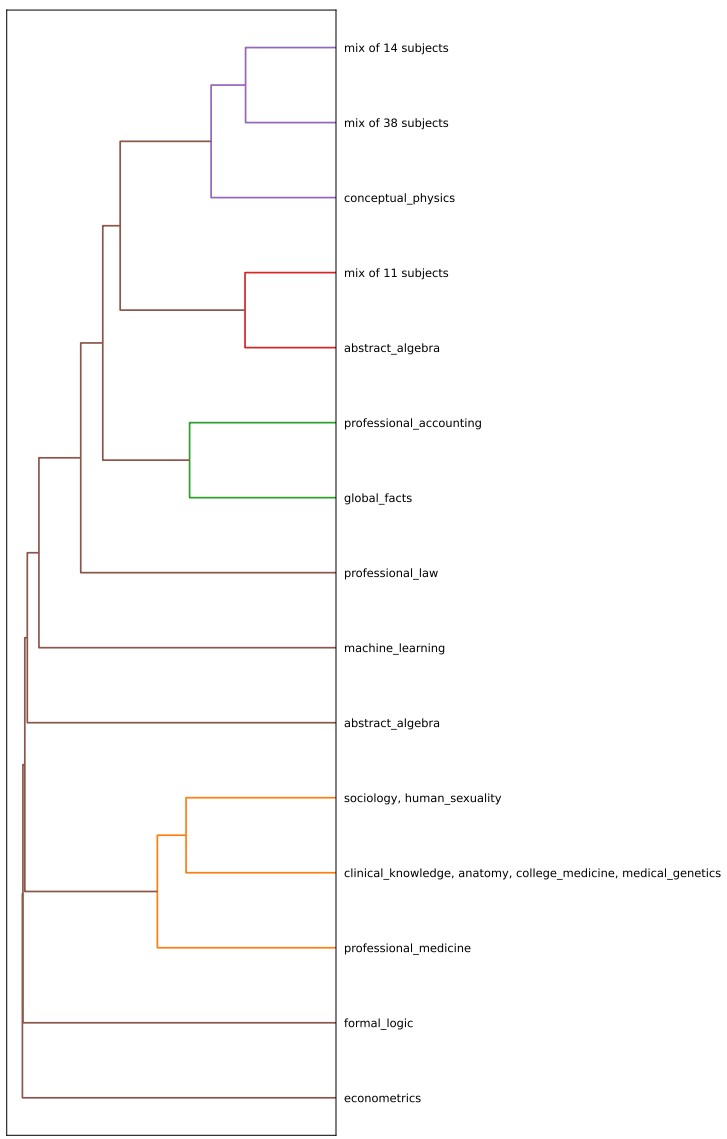

Figure A26: **Llama3-8b, finetuned, Z=4, ARI = 0.05, 17 clusters**

