# OpenReview forum: "The Representation Landscape of Few-Shot Learning and Fine-Tuning in Large Language Models"
_NeurIPS.cc/2024/Conference — NeurIPS 2024 poster_

### Official Review · Reviewer_UJWa · 2024-07-10

**Soundness:** 2
**Presentation:** 2
**Contribution:** 2
**Rating:** 5
**Confidence:** 3

**Summary:**

This work focuses on understanding supervised finie-tuning (denoted SFT, where parameter weights are changed) and in-context learning (denoted ICL, where weights remain frozen and the adaptivity is done through a few shots in prompts), in the context of modern LLM. In doing so, the authors propose to apply the Advanced Density Peaks algorithm (ADP) to layers in the transformers that are used in a question answering task. The authors presented a wide range of plots and discussion, observing that both few-shot learning and supervised fine-tuning shows complementary behavior where earlier layers and late layers focus on different levels of information.

**Strengths:**

This manuscript could gain importance due to the recent advances in modern LLMs and the widely adapted practice of supervised fine-tuning and  in-context learning. The authors make great efforts showing a wide range of analysis (although in a less clearly organized manner).

**Weaknesses:**

I have a few major concerns regarding the manuscript in its current form.

This manuscript has several issues in clarity. First, the description of the ADA algorithm, which is the cornerstone of this work, remains largely hard to follow, and I have to refer to the original ADA paper to figure out what the authors propose. Also in the results section, a wide range of observations are discussed, but such discussions are largely scattered and I found it hard to gain insight from them.

There are also issues with the originality and significance of this manuscript. This is partially due to the issues in clarity mentioned above and also the lack of significant contributions. Technically, this manuscript adapts ADA algorithms in understanding the representation, in each layer of the transformer. This is technically not a significant contribution. This, alone, is not a concern, since many great analysis works use simple methods to reveal important insight. However, in this manuscript’s case, the insights are less clear and a reader may find it difficult to draw inspiration from the discussions.

In an ideal form, the manuscript could contain clear, organized discussion revealing interesting insights. A non-exhausting topic that can be studied includes semantics, fine-tuning dynamics, theoretical bounds, practical rule-of-thumb, or the lack thereof in modern models. I would encourage the authors to consider such aspects.

Nit-picketing:
Table A1: last columns for Llama-3-70b have formatting issues (should uniformly use percentage or decimal).

**Questions:**

N/A

**Limitations:**

Yes.

---

> ### Author Rebuttal · Authors · 2024-08-06
>
> Thank you for the time spent reading our manuscript and the concerns you raised, which will help improve our study and make its exposition clearer. We will address all your points below.
>
> > *This manuscript has several issues in clarity. [...]*
>
> We agree with you that the description of the ADP algorithm can be improved, and we plan to do so in the revised version of the paper. We plan to modify the ADP exposition as follows:
> 1. To facilitate understanding of the algorithm's logical flow, we will split the description of the ADP algorithm into four paragraphs: “intrinsic dimension estimation”, “density estimation,” “density-based clustering,” and “hierarchical clustering of the peaks with WPGMA.”
> 2. We will focus on the algorithm's assumptions and give an intuitive picture, adding a figure to represent the geometrical intuition behind the ADP.
> 3. We will include all the technical details in the appendix (we already did that in Sec. B) to avoid making the exposition in the main paper too long and difficult to follow.
>
> >*discussions are largely scattered [...]*
>
> The main messages of our work, as explicitly stated in the bullet points of the introduction, are the following:
> 1. Across the hidden layers of LLMs, the probability landscape changes in a two-phased manner, separated by a sharp transition in the middle of the network (1st bullet point, and Sec 3.1)
> 2. When LLMs are adapted to solve a downstream task, ICL and SFT exhibit complementary behavior with respect to these two phases: \
> 2a. before the transition, ICL induces an interpretable organization of the density peaks, which is consistent with the semantic information of the data (2nd bullet point, and Sec 3.2) \
> 2b. SFT modified the geometry of the probability density after the transition where few (3rd bullet point, and Sec 3.3)
>
> In the revised version of the manuscript, we will make this correspondence clearer by reducing the bullet points from 4 to 3 so that they can explicitly match each section of the results and assign more explicit titles to Sec. 3.2 and 3.3.
>
> > *There are also issues with the originality [...]*
>
> We believe that our study has a number of methodological and substantial novel contributions: \
> While most of the previous works compare ICL and finetuning in terms of performance on downstream tasks (see e.g. [1] and refs. therein), we propose a very different analysis of the changes in the geometry of the hidden layers induced by these two paradigms;\
> While most previous studies trained linear probes to discover semantic information encoded in hidden layers (see [2-3] among others), we propose and use a fully unsupervised nonparametric approach to study the probability landscape in LLM, finding interpretable semantic relations between the modes of distribution;\
> Finally, we validate our claims on state-of-the-art models like Llama-3, released only a few months ago (April 2024), including a study of the 70B parameter models, which are often overlooked.
>
> >*Topics that can be studied include semantics, fine-tuning dynamics, theoretical bounds, practical rule-of-thumb [...].*
>
> We appreciate your perspective on this matter. However, we'd like to respectfully clarify that our work does indeed address the *semantics* and *fine-tuning dynamics* in modern models. In addition, these analyses allow us to give clear-cut practical rules-of-thumb.
>
> *Semantics*:\
> In Section 3.2, we explore the extent to which the contextual embeddings encode a global property of prompts, specifically the topic of the question. We show that in ICL, the nearby density peaks contain the embeddings of
> sentences denoting similar topics (e.g. math, physics, ...), and their global hierarchical organization closely mirrors the semantic similarity of the subjects (see lines 182-185, 212-214, 229-236, and Figure 4).
> In line 44 and bullet points 1, 2, and 4, we explicitly state that our work also addresses semantic analysis.
>
> *Fine-tuning dynamics*:\
> Figure 6 and lines 275-284 are dedicated to the analysis of the fine-tuning dynamics through the lens of the representation similarity between hidden layers before and after fine-tuning.
> This analysis shows that most training steps are spent chaining the last layers rather than the early ones (see lines 282-284).
> In Figure 3 of the accompanying PDF, we show that as the late layers become more dissimilar to their initial state, their ARI with the letters correspondingly increases. This change results from the onset of a few homogeneous density peaks on the answer label. \
> This analysis addresses another point raised by the reviewer, namely, the discussion of practical rules of thumb.
>
> *Practical rule-of-thumb*:\
> Since fine-tuning affects the second half of the network, our analysis suggests a possible strategy to adjust the ranks of the LoRA matrices. Several studies [4-6] attempt to adapt the ranks of the LoRA matrices based on different notions of matrix ‘importance.’ Our study of the dynamics of similarity (Fig. 6) gives a principled measure of importance: the layers that change the most should have higher ranks. We are currently investigating these aspects in our lab and will include a brief discussion of the practical application of our study in the revised version of the manuscript.
>
> We thank you again for pointing out clarity issues since these will allow us to improve our manuscript. We hope our responses can alleviate your concerns regarding the clarity of our results. If so, we hope you will consider raising your score.
> We’ll be happy to reply to any further questions.
>
> [1] Mosbach et al., Few-shot Fine-tuning vs. In-context Learning: A Fair Comparison and Evaluation\
> [2] Conneau et al, What you can cram into a single vector: Probing sentence embeddings for linguistic properties\
> [3] Hewitt and Liang, Designing and interpreting probes with control tasks.\
> [4] Zhang et al., AdaLoRA\
> [5] Valipour et al., DyLoRA\
> [6] Ding et al., Sparse Low-rank Adaptation of Pre-trained Language Models.

---

> ### Comment · Reviewer_UJWa · 2024-08-13
>
> Thanks for the detailed rebuttal.
>
> Having read the response and the conversation between the authors and other reviewers, I'm happy that the authors propose revisions to deal with some issues in clarity. Also I'm glad that the extensive set of experiments have been well recognized by all reviewers. However, a few issues I'm concerning still exists, especially on how scatters observations are grouped and discussed in a coherent and concise way -- which is important for readers in the community to gain useful insights. As a result, I revised my score to reflect the positive side of the authors' articulation in rebuttal, with the hope that the promised revision could be realized in full extent.

---

> > ### Author Response · Authors · 2024-08-13
> >
> > We thank you for acknowledging the extensive set of experiments that support the main message of our contribution and for increasing  your score.
> >
> > We take your concerns on clarity seriously and are committed to include the promised improvements in the camera ready version of this work.

---

### Official Review · Reviewer_mxnJ · 2024-07-12

**Soundness:** 3
**Presentation:** 3
**Contribution:** 3
**Rating:** 6
**Confidence:** 3

**Summary:**

The paper explores the internal dynamics of LLMs when subjected to ICL and SFT strategies. Despite achieving comparable outcomes, the paper reveals that these methods result in distinct representation landscapes within the model. ICL fosters a hierarchical organization of representations based on semantic content in the initial layers, whereas SFT yields a more diffuse and semantically mixed landscape. In the latter stages of the model, fine-tuning leads to representations with sharper probability modes that better capture answer identities compared to the less defined peaks produced by ICL.

**Strengths:**

The paper takes a novel approach by analyzing the geometric properties of the hidden representations in LLMs, offering new insights into the workings of ICL and SFT. The use of the Advanced Density Peaks algorithm and the systematic investigation of data transformations across hidden layers demonstrate a insightful methodological framework.

**Weaknesses:**

See Questions.

**Questions:**

1. Do few-shot examples significantly impact the analysis results? If so, in what aspects are these impacts manifested?
2. Is te method applicable to studying tasks that generate longer texts?
3. What do you think the specific practical applications of the findings might be? For example, improving ICL or SFT methods, or perhaps combining the two in some way?

---

> ### Author Rebuttal · Authors · 2024-08-06
>
> Thank you for the time spent reading our manuscript, for your general appreciation of our work, and for your questions. We will answer point by point below.
>
> > *1. Do few-shot examples significantly impact the analysis results? If so, in what aspects are these impacts manifested?*
>
> The order and identity of the shots can slightly affect few-shot accuracy, it has a smaller impact on the clusters found by the ADP algorithm and, more generally, on our results. In Table A1 in the appendix of the main paper, we show that the 5-shot accuracy can change from 66.7 to 66.2, changing the few-shot order in Llama 3 8b.
>
> We performed an additional experiment to test the impact of the order and identity of the shots on the ARI with the subjects. Figure 3 in the attached pdf shows the results of this test. We measured the standard deviation of the ARI with 5 runs with different shot orders by shuffling the MMLU dev set (red profile) and the identity of the few shots sapling from the union of the dev and validation set. On average, the standard deviation is about $\sim0.02$. This means that our findings are robust to changes in the prompt since the number of clusters and their internal composition remain consistent across different runs.
>
> > *2. Is the method applicable to studying tasks that generate longer texts?*
>
> The method can be applied to sequences with more than one token. However, since the clustering and intrinsic dimension estimation rely on the Euclidean distances between samples, it is important to map the generated sequence to a common embedding.
> One strategy is to average over the sequence tokens as done, for instance, by Valeriani et al. [1] who measure the intrinsic dimension of biological sequences of varying length by averaging over the sequence axis.
>
> > *3. What do you think the specific practical applications of the findings might be? For example, improving ICL or SFT methods, or perhaps combining the two in some way?*
>
> Our findings can help improve strategies for LoRA finetuning that adapt the rank for LoRA matrices. Several studies [2-4] attempt to adapt the ranks of the LoRA matrices based on various notions of matrix ‘importance’ or relevance to downstream tasks.
> Our analysis of the similarity between finetuned layers and the pre-trained layers (see Fig. 6 and lines 275 284) shows that late layers are most significant for finetuning. This indicates that the layers that change the most should have higher ranks. This approach would naturally prevent the early layers from being modified by fine-tuning and could combine the benefits of ICL and FT, as you correctly suggest.
> We are actively working on these aspects in our lab and will include a brief discussion of the practical application of our study in the revised version of the manuscript.
>
> [1] Valeriani et al., The geometry of hidden representation in Large Transformer Models \
> [2] Zhang et al., AdaLoRA: Adaptive Budget Allocation for Parameter-Efficient Fine-Tuning.\
> [3] Valipour et al., DyLoRA: Parameter Efficient Tuning of Pre-trained Models using Dynamic Search-Free Low-Rank Adaptation.\
> [4] Ding et al., Sparse Low-rank Adaptation of Pre-trained Language Models

---

> > ### Comment · Reviewer_mxnJ · 2024-08-13
> >
> > Thank you for your response. I am interested in the topic of your paper and recognize its contribution to the community. However, I share the concerns of other reviewers about the need for improved clarity and organization in the writing, as well as the shortcomings in the research novelty. These considerations have led me to adjust my score accordingly.

---

> > > ### Author Response · Authors · 2024-08-13
> > >
> > > We thank the reviewer for the response. \
> > > Since we believe that our work has several novel contributions we take the occasion to summarize them further below:
> > > 1. we compare ICL and fine-tuning by analyzing the *geometry of the layers* rather than using performance metrics on downstream tasks;
> > > 2. we describe the geometry of the representations with the *density peaks clustering*. This fully unsupervised nonparametric algorithm harnesses the low dimensional structure of the data to interpret representations in the hidden layers. Our work is the first to effectively employ this strategy to interpret the content of hidden layers since most of the previous studies use supervised probes;
> > > 3. we apply this powerful methodological framework to *analyze SOTA LLMs* (e.g. llama3, ...), we show that it *scales to 70B* parameter models, and it can extract clear, interpretable patterns from the hidden representations of such models;
> > > 4. we show that the geometry of the hidden layers changes abruptly in the middle of the models. ICL and fine-tuning modify the geometry of layers' representations in radically different regions of the LLMs: ICL before and fine-tuning after the transition observed in the middle of the LLMs. We are the first to observe and interpret this different behavior of ICL and fine-tuning inside LLMs.\
> > > For these reasons, we believe that our contribution has elements of novelty in its perspective (1), methodology (2-3), and findings (4).
> > >
> > > Part of the clarity issues concerns the exposition of the density peaks clustering and WPGMA linkage strategy. \
> > > We did not develop these algorithms in this work, but for the first time we employ them to analyze LLM representations, showing that they can be powerful tools for discovering new, insightful findings.
> > > We cited the original papers where these techniques were developed; thanks to the concerns raised by the reviewers we are committed to improving further their exposition in the camera-ready version of this work.

---

### Official Review · Reviewer_P3MS · 2024-07-13

**Soundness:** 3
**Presentation:** 3
**Contribution:** 3
**Rating:** 6
**Confidence:** 4

**Summary:**

In this paper, the authors analyze the probability landscapes of the hidden representations of LLMs when they perform in-context learning (ICL) and supervised fine-tuning (SFT) using the MMLU question answering benchmark, and the Llama-3, Llama-2, and Mistral-7B LLMs are used for this purpose. The Advanced Density Peaks (ADP) algorithm is used for this analysis. The experiments conducted recover fingerprints that showcase the distinctions between ICL and SFT, specifically that representations in the earlier layers of the network are aligned with the subjects (topic of the question) during ICL and that later layers are better aligned with the final answers for SFT. The authors conduct a number of different experiments analyzing the probability landscape and draw interesting conclusions from the results.

**Strengths:**

- The paper has a number of strengths, namely the extensive set of experiments that lead to interesting findings. The findings are useful to the community-- with the main takeaway being the two-phased nature of the hidden layer representations' probability landscapes.
- Additionally, I appreciate the findings obtained that demonstrate the fingerprints obtained for ICL and SFT in the early and later layers respectively. More specifically, the results imply that during ICL the representations in the earlier layers of the network are more aligned with the subject partitions and that during SFT representations of the later layers are better aligned with the answer/label distributions of questions. This finding showcases the distinctions between ICL and SFT from the perspective of the representation landscape.
- The experiments on the hierarchical organization of the cluster density peaks via linkage clustering with respect to the subject relationships show how information in the internal model layers are organized.

**Weaknesses:**

- My main issue with this work is that the experimental analysis is only conducted on the MMLU benchmark. This detrimentally restricts the impact of the work, especially as the authors draw general conclusions. Is it possible to extend some of this experimental analysis to other benchmarks, perhaps just even to other question answering based benchmarks?
- Can the authors provide more details on the average linkage experiments that demonstrate the hierarchical organization of the cluster density peaks? For instance, how do the authors map the subjects at a granular level to each of the dendograms/leaves? Overall, I believe the subsection "Hierarchical organization of density peaks" can be further improved in terms of writing and readability for future readers.
- There are a number of typos throughout the paper that need to be corrected. For example, the authors write "LLama" in a number of places although it should read "Llama" (e.g. line 86), "assignation" -> "assignment" on line 140, among others. These can be fixed in the revision.

**Questions:**

Please see the weaknesses listed above as each of those is also framed as a question to the authors.

**Limitations:**

The authors discuss limitations sufficiently. Note that owing to the strengths and weaknesses mentioned above, the paper's contributions currently constitute a technically solid, moderate-to-high impact work, which forms the basis for my score.

---

> ### Author Rebuttal · Authors · 2024-08-06
>
> We are grateful for your constructive and thoughtful comments and for the time spent on our manuscript. We will address all the main concerns below:
>
> > *Is it possible to extend some of this experimental analysis to other benchmarks?*
>
> In our work, we analyzed MMLU because it includes a wide variety of topics (57), with clear semantic relationships between them and a rich set of samples (more than 100) for each topic. These requirements are hard to find in other datasets.
>
> We agree with you that an additional benchmark would strengthen our claims. Therefore following your suggestion, we performed a new experiment on a second dataset constructed from TheoremQA [1], ScibenchQA [2], Stemez [3], and RACE [4]. This dataset contains roughly 6700 examples, not included in MMLU, 10 subjects from STEM topics [1-3], and middle school reading comprehension task [4], with at least 300 examples per subject. We keep 4 choices for each answer. The 0 shot, 5 shot, and fine-tuned accuracies in Llama 3 8b are 55%, 57%, and 58%, respectively.
> We report the analysis in Fig. 1 and 2 of the attached PDF. In Fig. 1-left, we see that the intrinsic dimension profiles have a peak around layers 15/16, the *same layers as in MMLU* (Fig. 2-left main paper). This peak in ID signals the transition between the two phases described in the paper.
> Before layer 17,  few-shot models encode better information about the subjects (ARI with subjects above 0.8). Between layers 3 and 7, the peaks in 5 shot layers reflect the semantic similarity of the subjects (see the dendrograms for layer 5 in Fig. 2 of the attached PDF).
>
> Fine-tuning instead changes the representations after layer 17, where ARI with the answers for the is $\sim 0.15$, higher than that of the 5 shot and 0 shot models. The absolute value is lower than that reported in the main paper (Fig. 5-left) because the fine-tuned accuracy reached on the STEM subjects in this dataset is lower.
>
> Overall, the results are consistent with those shown in the paper for MMLU.
>
> > *Can the authors provide more details on the average linkage experiments [...]? How do the authors map the subjects [...] to each of the dendograms/leaves?*
>
> The two key points to understanding our procedure are:
> 1. Each peak is associated with a single subject;
> 2. The saddle points between the density peaks (subjects) allow us to cluster them hierarchically using Weighted Pair Group Method with Arithmetic Mean (WPGMA)..
>
> We elaborate on these points below.
>
> 1. The clusters are homogeneous in the layers where ARI is highest (layers 4 to 6, Fig. 3-left of the main manuscript). In these layers in Llama 3 8b, 51 out of 77 clusters are pure, and in 70 out of 77, 90% of the points belong to the same subject. This high purity allows us to map one or two subjects to one density peak, which becomes a leaf of the dendrogram (see also Fig. A9).
>
> 2. Average linkage is a strategy to do hierarchical clustering on the density peaks. Hierarchical clustering requires a notion of **dissimilarity** between the peaks/subjects [5].  The Advanced Density Peaks (ADP) defines the dissimilarity between a pair of peaks $\alpha$ and $\beta$ as [6]: \
> $S_{\alpha, \beta} = \log \rho_{max} - \log \rho_{\alpha,\beta}$ \
> where $\rho_{max}$ is the density of the highest peak, and $\rho^{\alpha, \beta}$ is  saddle point density between $\alpha$ and $\beta$.
> Intuitively, the higher is $\log \rho^{\alpha, \beta}$, the smaller is $S_{\alpha, \beta}$. The ADP algorithm considers two peaks similar if they are connected through a high-density saddle point. \
> The peaks are then linked hierarchically, starting from the pair with the lowest dissimilarity according to $S_{\alpha, \beta}$.  Once two peaks $\alpha$, $\beta$ have been merged into a peak \gamma, the saddle point density between the peak representing their union and the rest of the peaks ${\delta_i}$ is determined according to the average linkage strategy (WPGMA) [7] (see line. 222) namely: \
> $ \log \rho^{\gamma, \delta_i}= \dfrac{\log \rho^{\alpha, \delta_i}+\log \rho^{\beta, \delta_i}}{2} $ \
> After the update of $S$ we repeat the procedure until we reach a single cluster which is the root of the dendrogram.
>
> We will include the description of the WPGMA methodology in the method section at the end of the section devoted to the ADP algorithm.
>
> > *There are a number of typos.*
>
> Thank you. we will fix them in the camera-ready version.
>
> We hope that our additional experiment addresses your main request and that we have satisfactorily clarified your second concern. If so, we hope you can consider raising your score.
> We'll be more than happy to give further replies to any remaining doubts.
>
>
> [1] Chen et. al, Theoremqa: A theorem-driven question answering dataset \
> [2] Wang et. al, Scibench: Evaluating college-level scientific problem-solving abilities of large language models. \
> [3] https://stemez.com/subjects \
> [4] Lai et al, RACE: Large-scale ReAding Comprehension Dataset From Examinations \
> [5] Hastie et al., Elements of statistical learning. \
> [6] D’Errico et al., Automatic topography of high-dimensional data sets by non-parametric density peak clustering. \
> [7] Sokal, A statistical method for evaluation systematic relationship.

---

> > ### Comment · Reviewer_P3MS · 2024-08-11
> >
> > I would like to thank the authors for their rebuttal. After going through it, I believe the paper's contributions still currently constitute a technically solid, moderate-to-high impact work as I had mentioned in my original review. I will hence keep my score. I would also request the authors to include the additional details on the WPGMA framework in the revision.

---

> > > ### Author Response · Authors · 2024-08-11
> > >
> > > Thank you for keeping your positive feedback about our work.
> > >
> > > We will include the details of the WPGMA algorithm in the revision.

---

### Author Rebuttal · Authors · 2024-08-06

We thank the chair and the senior area chair for their time spent reviewing our work.

*Summary of our contribution*:\
In this study, we compare fine-tuning (FT) and in-context learning (ICL) in LLMs by
analyzing the evolution of the probability density in the hidden layers of state-of-the-art LLMs (Llama-3 8B/70B, Llama-2 7B/13B/70B, Mistral 7B) as they solve a challenging question answering task (MMLU). \
We find that the evolution of the probability density in LLMs occurs in two phases, separated by a sharp modification of the representation landscape in the middle layers. In early layers, ICL promotes a remarkable organization of the probability modes that are close to each other according to the semantic similarity of the MMLU subjects. In contrast, FT  affects the probability landscape of late layers, leaving that of the early layers mostly unchanged.

*Summary of reviewers’ praise and concerns*:\
Reviewer **mxnJ** appreciated the novelty of our approach, which analyzes *the geometric properties of the hidden representations in LLMs, offering new insights into the workings of ICL and SFT*. For Reviewer **P3MS** our analysis of the representation landscape *has a number of strengths, namely the extensive set of experiments that lead to interesting findings useful to the community*. **UJWa** acknowledges the *extensive set of experiments* carried out in our work.\
The reviewers' main concerns were:
1. Lack of clarity in some parts of our exposition (**UJWa** and **P3MS**).
2. The need for an additional dataset to strengthen our claims (**P3MS**).

*Summary of our reply to reviewers' concerns*:
1. We have addressed reviewer **P3MS**'s primary concern with a new experiment on a *second dataset*, which confirms and strengthens our findings (see attached pdf Figures 1 and 2).
2. To improve the clarity of our exposition, we will: \
2a. Organize the description of the Advanced Density Peaks algorithm into four distinct paragraphs (see reply to **UJWa**), including an explicit description of the average linkage algorithm we use (see reply to **P3MS**).\
2b. We will make explicit the match between bullet points and the section of the Results (see reply to **UJWa**) and polish the paragraph about the Hierarchical organization of the density peaks, as suggested by Reviewer **P3MS**.

We hope that our additional experiment and our replies to the reviewers' comments address their concerns. We are open to clarifying any remaining questions.

---

### Decision · Program_Chairs · 2024-09-25

**Decision:**

Accept (poster)

**Comment:**

This work studies the geometry of the representations of large language models, comparing the behaviors observed early and late in the models in two settings, for fine-tuned models and for in-context learning. The authors perform a solid empirical study and produce some quite interesting findings (e.g., ICL produces certain hierarchical behaviors that resemble the relationships between semantic concepts).

Reviewers are generally positive, and I agree with them. This is a nice work that tries to better understand LLM functionality for popular approaches rather than focusing on new approaches or techniques, something that is definitely needed in this space.